# VARIATIONAL INTRINSIC CONTROL REVISITED

**Taehwan Kwon**
NC
kth315@ncsoft.com

## ABSTRACT

In this paper, we revisit variational intrinsic control (VIC), an unsupervised reinforcement learning method for finding the largest set of intrinsic options available to an agent. In the original work by Gregor et al. (2016), two VIC algorithms were proposed: one that represents the options explicitly, and the other that does it implicitly. We show that the intrinsic reward used in the latter is subject to bias in stochastic environments, causing convergence to suboptimal solutions. To correct this behavior and achieve the maximal empowerment, we propose two methods respectively based on the transitional probability model and Gaussian mixture model. We substantiate our claims through rigorous mathematical derivations and experimental analyses.

## 1 INTRODUCTION

Variational intrinsic control (VIC) proposed by Gregor et al. (2016) is an unsupervised reinforcement learning algorithm that aims to discover as many *intrinsic* options as possible, i.e., the policies with a termination condition that meaningfully affect the world. The main idea of VIC is to maximize the mutual information between the set of options and final states, called *empowerment*. The maximum empowerment is desirable because it maximizes the information about the final states the agent can achieve with the available options. These options are independent of the extrinsic reward of the environment, so they can be considered as the agent's universal knowledge about the environment.

The concept of empowerment has been introduced in (Klyubin et al., 2005; Salge et al., 2014) along with methods for measuring it based on Expectation Maximization (Arimoto, 1972; Blahut, 1972). They defined the option as a sequence of a fixed number of actions. Yeung (2008) proposed to maximize the empowerment using the Blahut & Arimoto (BA) algorithm, but its complexity increases exponentially with the sequence length, rendering it impractical for high dimensional and long-horizon options. Mohamed & Rezende (2015) adopted techniques from deep learning and variational inference (Barber & Agakov, 2003) and successfully applied empowerment maximization for high dimensional and long-horizon control. However, this method maximizes the empowerment over *open-loop* options, meaning that the sequence of action is chosen in advance and conducted regardless of the (potentially stochastic) environment dynamics. This often impairs the performance, as the agent cannot properly react to the environment, leading to a significant underestimation of empowerment (Gregor et al., 2016).

To overcome this limitation, Gregor et al. (2016) proposed to use the *closed-loop* options, meaning that the sequence of action is chosen considering the transited states. This type of options differ from those in Klyubin et al. (2005), Salge et al. (2014) and Mohamed & Rezende (2015) in that they have a termination condition, instead of a fixed number of actions. They presented two algorithms: VIC with explicit and implicit options (we will call them *explicit* and *implicit VIC* from here on). Explicit VIC defines a fixed number of options, and each option is sampled at the beginning of the trajectory, conditioning the policies of an agent until termination. In other words, both the state and the sampled option are the input to the policy function of the agent. One clear limitation of explicit VIC is that it requires the preset number of options. This does not only apply to explicit VIC, but also to some recent unsupervised learning algorithms that adopt a discrete option or skill with a predefined set (Machado et al., 2017; Eysenbach et al., 2018). Obviously, presetting the number of options limits the number of options that an agent can learn, impeding the maximal level of empowerment. Moreover, choosing a proper number of options is not straightforward, since the maximum of the objective for a given number of options depends on several unknown environmental factors such as

the cardinality of the state space and the transitional model. To overcome this issue, Gregor et al. (2016) proposed implicit VIC which defines the option as the trajectory (i.e., the sequence of states and actions) until termination. There exist multiple trajectories that lead to the same final state and implicit VIC learns to maximize the mutual information between the final state and the trajectory by controlling the latter. As a result, the number of options is no longer limited by the preset number, and neither is the empowerment. Despite this advantage, however, these implicit options induce bias in the intrinsic reward and hinder implicit VIC from achieving maximal empowerment. This effect grows with the stochasticity of the environment, and it may cause serious degradation of empowerment, limiting the agent from learning the universal knowledge of the environment. In this paper, we aim to solve such empowerment degradation of implicit VIC under stochastic dynamics. To this end, we revisit variational intrinsic control and make the following contributions:

1. We show that the intrinsic reward in implicit VIC suffers from the variational bias in stochastic environments, causing convergence to suboptimal solutions (Section 2).
2. To compensate this bias and achieve the maximal empowerment, we suggest two modifications of implicit VIC: the environment dynamics modeling with the transitional probability (Section 3) and Gaussian mixture model (Section 4).

## 2 VARIATIONAL BIAS OF IMPLICIT VIC IN STOCHASTIC ENVIRONMENTS

In this section, we derive the variational bias of implicit VIC under stochastic environment dynamics. First, we adopt the definition of termination action and final state from Gregor et al. (2016) and briefly review VIC. The termination action terminates the option and yields the final state $s_f = s_t$ independently of the environmental action space. VIC aims to maximize the empowerment, i.e., the mutual information between option $\Omega$ and final state $s_f$, which can be written as follows:

$$I(\Omega, s_f|s_0) = -\sum_{\Omega} p(\Omega|s_0) \log p(\Omega|s_0) + \sum_{\Omega, s_f} p(s_f|s_0, \Omega) p(\Omega|s_0) \log p(\Omega|s_0, s_f). \quad (1)$$

Since $p(\Omega|s_0, s_f)$ is intractable, VIC (Gregor et al., 2016) derives the variational bound $I^{VB} \leq I$ and maximizes it instead:

$$I^{VB}(\Omega, s_f|s_0) = -\sum_{\Omega} p(\Omega|s_0) \log p(\Omega|s_0) + \sum_{\Omega, s_f} p(s_f|s_0, \Omega) p(\Omega|s_0) \log q(\Omega|s_0, s_f), \quad (2)$$

where $q(\Omega|s_0, s_f)$ is the inference model to be trained. When $I^{VB}$ is maximized, we have $q(\Omega|s_0, s_f) = p(\Omega|s_0, s_f)$ and achieve the maximum $I$. As explained in Section 1, explicit VIC samples an explicit option at the beginning of a trajectory and it conditions policy as $\pi(a|s, \Omega)$ until termination. Due to the randomness of policy, the final state is undetermined for a given option until the policy converges to achieve a specific option. Unlike explicit VIC, implicit VIC defines its option as a trajectory until termination, and hence, the final state is determined for a given option. This can be expressed as

$$p(s_f|\Omega, s_0) = \begin{cases} 1, & \text{if } s_f = s_{f|\Omega} \\ 0, & \text{otherwise} \end{cases} \quad (3)$$

where $s_{f|\Omega}$ is the final state of an option $\Omega$. This is a key characteristic of implicit VIC and is essential for deriving the main results of this paper. We will often use this equation to eliminate $s_f$ for reduction. Interestingly, this makes a difference in empowerment maximization between explicit and implicit VIC, which can be explained by rewriting $I(\Omega, s_f|s_0)$ as follows:

$$I(\Omega, s_f|s_0) = H(s_f|s_0) - H(s_f|\Omega, s_0).$$

Note that $H(s_f|\Omega, s_0)$ is 0 for implicit VIC since $s_f$ is determined for a given $\Omega$. One can notice that to maximize empowerment, the agent needs to learn (1) maximizing $H(s_f|s_0)$ and (2) minimizing $H(s_f|\Omega, s_0)$. While explicit VIC needs to learn both (1) and (2), implicit VIC needs to learn only (1), since (2) is already achieved by the definition of option. This makes the learning of implicit VIC easier and faster. Moreover, implicit VIC is scalable as explained in Section 1. Despite these strengths, implicit VIC suffers from variational bias in intrinsic reward under stochastic environments that can seriously degrade the empowerment. We derive this variational bias by decomposing $p(\Omega|s_0)$ and $p(\Omega|s_f, s_0)$. Since implicit VIC defines the option $\Omega$ as the trajectory of an agent, i.e.,

the sequence of states and actions: $\Omega = (s_0, a_0, s_1, a_1, ..., s_{T-1}, a_{T-1} = a_f, s_T = s_f = s_{T-1})$, the probability of an option can be decomposed as

$$p(\Omega|s_0) = \prod_{(\tau_t, a_t, s_{t+1}) \in \Omega} p(a_t|\tau_t) p(s_{t+1}|\tau_t, a_t) \tag{4}$$

using Bayes rule with $\tau_t = (s_0, a_0, ..., s_t)$. Similarly, $p(\Omega|s_0, s_f)$ can be expressed as

$$p(\Omega|s_0, s_{f|\Omega}) = \prod_{(\tau_t, a_t, s_{t+1}) \in \Omega} p(a_t|\tau_t, s_{f|\Omega}) p(s_{t+1}|\tau_t, a_t, s_{f|\Omega}). \tag{5}$$

Note that $s_f$ is replaced by $s_{f|\Omega}$ since it is determined by the given $\Omega$. Using (4), (5) and (3), we can rewrite the mutual information (1) of implicit VIC as

$$I(\Omega, s_f|s_0) = \sum_{\Omega} p(\Omega|s_0) \sum_{(\tau_t, a_t, s_{t+1}) \in \Omega} \left[ \log \frac{p(a_t|\tau_t, s_{f|\Omega})}{p(a_t|\tau_t)} + \log \frac{p(s_{t+1}|\tau_t, a_t, s_{f|\Omega})}{p(s_{t+1}|\tau_t, a_t)} \right]. \tag{6}$$

The intrinsic reward of implicit VIC (Gregor et al., 2016) is given by

$$r_{\Omega}^{I^{VB}} = \sum_{(\tau_t, a_t, s_{t+1}) \in \Omega} \log \frac{q(a_t|\tau_t, s_{f|\Omega})}{p(a_t|\tau_t)}, \tag{7}$$

where $q(a_t|\tau_t, s_f)$ is inference and $p(a_t|\tau_t)$ is policy of an agent. It can be shown that $r_{\Omega}^{I^{VB}}$ comes from the first part of (6) (see Appendix A for details). Under deterministic environment dynamics, the transitional part $\log p(s_{t+1}|\tau_t, a_t, s_{f|\Omega})/p(s_{t+1}|\tau_t, a_t)$ is canceled out since both the nominator and denominator are always 1. However, this is not possible under stochastic environment dynamics and it yields the variational bias $b_{\Omega}^{VIC}$ in the intrinsic reward:

$$b_{\Omega}^{VIC} = \sum_{(\tau_t, a_t, s_{t+1}) \in \Omega} \log \frac{p(s_{t+1}|\tau_t, a_t, s_{f|\Omega})}{p(s_{t+1}|\tau_t, a_t)}. \tag{8}$$

From (8), we see that this bias comes from the difference between the transitional probabilities with and without the given final state. This difference is large when $s_{f|\Omega}$ in the nominator plays a crucial role, which then causes a large bias. One extreme case is when $s_{t+1}$ is the necessary transition to reach $s_{f|\Omega}$ but not the only transition from $(\tau_t, a_t)$. Then, $p(s_{t+1}|\tau_t, a_t, s_{f|\Omega})$ is 1 but $p(s_{t+1}|\tau_t, a_t)$ is not, yielding a large bias. In Section 5, we provide the experimental evidence that this variational bias leads to a suboptimal training. Even though the original VIC (Gregor et al., 2016) subtracts $b(s_0)$ from $r_{\Omega}^{I^{VB}}$ to reduce the variance of learning, it cannot compensate this bias since it also depends on $\Omega$. In the next section, we analyze the mutual information (1) in more detail under stochastic environment dynamics and define the variational estimate of (1), $I^{VE}$, for training.

## 3   IMPLICIT VIC WITH TRANSITIONAL PROBABILITY MODEL

In this section, we analyze $I(\Omega, s_f|s_0)$ under stochastic environment dynamics and propose to explicitly model transitional probabilities to fix variational bias. First, for a given option and final state, we define $p_\pi(\Omega|s_f, s_0)$, $p_\rho(\Omega|s_f, s_0)$, $p_\pi(\Omega|s_0)$ and $p_\pi(\Omega|s_0)$ as follows:

$$\begin{aligned} p_\pi(\Omega|s_f, s_0) &= \prod_{(\tau_t, a_t, s_{t+1}) \in \Omega} p_\pi(a_t|\tau_t, s_f), \quad p_\rho(\Omega|s_f, s_0) = \prod_{(\tau_t, a_t, s_{t+1}) \in \Omega} p_\rho(s_{t+1}|\tau_t, a_t, s_f), \\ p_\pi(\Omega|s_0) &= \prod_{(\tau_t, a_t, s_{t+1}) \in \Omega} p_\pi(a_t|\tau_t), \qquad p_\rho(\Omega|s_0) = \prod_{(\tau_t, a_t, s_{t+1}) \in \Omega} p_\rho(s_{t+1}|\tau_t, a_t) \end{aligned} \tag{9}$$

Note that $p(\Omega|s_f, s_0) = p_\pi(\Omega|s_f, s_0) p_\rho(\Omega|s_f, s_0)$ is the true distribution of $\Omega$ for given $s_f$ where $p_\pi(\Omega|s_f, s_0)$ is a policy-related part and $p_\rho(\Omega|s_f, s_0)$ is a transitional part of $p(\Omega|s_f, s_0)$ and so do $p(\Omega|s_0)$, $p_\pi(\Omega|s_0)$ and $p_\rho(\Omega|s_0)$. It is necessary to consider transitional probabilities since they induce variational bias in intrinsic reward. Hence we model (9) as follows:

$$\begin{aligned} \pi^q(\Omega|s_f, s_0) &= \prod_{(\tau_t, a_t, s_{t+1}) \in \Omega} \pi^q(a_t|\tau_t, s_f), \quad \rho^q(\Omega|s_f, s_0) = \prod_{(\tau_t, a_t, s_{t+1}) \in \Omega} \rho^q(s_{t+1}|\tau_t, a_t, s_f), \\ \pi^p(\Omega|s_0) &= \prod_{(\tau_t, a_t, s_{t+1}) \in \Omega} \pi^p(a_t|\tau_t), \qquad \rho^p(\Omega|s_0) = \prod_{(\tau_t, a_t, s_{t+1}) \in \Omega} \rho^p(s_{t+1}|\tau_t, a_t), \end{aligned} \tag{10}$$

where $\pi^q$, $\rho^q$, $\pi^p$ and $\rho^p$ are our models to be trained. We know the policy of an agent, so we have $p_\pi(a_t|\tau_t) = \pi^p(a_t|\tau_t)$. For the other probabilities, they are trained based on our algorithms. Now we can rewrite $I(\Omega, s_f|s_0)$ as below using (9):

$$I(\Omega, s_f|s_0) = \sum_{\Omega, s_f} p(\Omega, s_f|s_0)\Big[\log p_\rho(\Omega|s_f, s_0)p_\pi(\Omega|s_f, s_0) - \log p_\rho(\Omega|s_0)p_\pi(\Omega|s_0)\Big]. \quad (11)$$

Using (10), we define $I^{VE}$ as follows:

$$I^{VE}(\Omega, s_f|s_0) = \sum_{\Omega, s_f} p(\Omega, s_f|s_0)\Big[\log \rho^q(\Omega|s_f, s_0)\pi^q(\Omega|s_f, s_0) - \log \rho^p(\Omega|s_0)\pi^p(\Omega|s_0)\Big]. \quad (12)$$

This is an estimate of the mutual information between $\Omega$ and $s_f$ with transitional models. Note that $I^{VE}$ is not a variational lower bound on $I$ unlike $I^{VB}$, hence the derivation of VIC in Gregor et al. (2016) is not applicable in this case. To tackle this problem, we start from absolute difference, $|I - I^{VE}|$ which can be bounded as

$$\Big|I - I^{VE}\Big| \le U^{VE} = \sum_{s_f} p(s_f|s_0)D_{\mathrm{KL}}\Big[p_\pi(\cdot|s_f, s_0)p_\rho(\cdot|s_f, s_0)\|\pi^q(\cdot|s_f, s_0)\rho^q(\cdot|s_f, s_0)\Big]$$
$$+ D_{\mathrm{KL}}\Big[p_\pi(\cdot|s_0)p_\rho(\cdot|s_0)\|\pi^p(\cdot|s_0)\rho^p(\cdot|s_0)\Big]. \quad (13)$$

See Appendix B for the derivation. Note that $U^{VE}$ is the sum of positively weighted KL divergences, which means that it satisfies $U^{VE} \to 0$ if and only if $\pi^q(\cdot|s_f, s_0) \to p_\pi(\cdot|s_f, s_0)$, $\rho^q(\cdot|s_f, s_0) \to p_\rho(\cdot|s_f, s_0)$ and $\rho^p(\cdot|s_0) \to p_\rho(\cdot|s_0)$ for all $s_f$. In other words, our estimate of the mutual information converges to the true value as our estimates (10) converge to the true distribution (9). It makes sense that we can estimate the true value of the mutual information if we know the true distribution. Hence we minimize $U^{VE}$ with respect to $\pi^q$, $\rho^q$ and $\rho^p$. If $\pi^q$, $\rho^q$ and $\rho^p$ are parameterized by $\theta_\pi^q$, $\theta_\rho^q$ and $\theta_\rho^p$, we can obtain gradients of $U^{VE}$ using (3) as follows:

$$\nabla_{\theta_\pi^q} U^{VE} = -\sum_{\Omega} p(\Omega|s_0)\nabla_{\theta_\pi^q} \log \pi^q(\Omega|s_{f|\Omega}, s_0),$$

$$\nabla_{\theta_\rho^q} U^{VE} = -\sum_{\Omega} p(\Omega|s_0)\nabla_{\theta_\rho^q} \log \rho^q(\Omega|s_{f|\Omega}, s_0), \quad (14)$$

$$\nabla_{\theta_\rho^p} U^{VE} = -\sum_{\Omega} p(\Omega|s_0)\nabla_{\theta_\rho^p} \log \rho^p(\Omega|s_0),$$

which can be estimated from sample mean. Once we have $(\pi^q(\cdot|s_f, s_0), \rho^q(\cdot|s_f, s_0), \rho^p(\cdot|s_0)) \approx (p_\pi(\cdot|s_f, s_0), p_\rho(\cdot|s_f, s_0), p_\rho(\cdot|s_0))$ for all $s_f$, we can update the policy to maximize $I$. If $\pi^p$ is parameterized by $\theta_\pi^p$, the gradients, $\nabla_{\theta_\pi^p} I$ and $\nabla_{\theta_\pi^p} I^{VE}$ can be obtained as follows using (3):

$$\nabla_{\theta_\pi^p} I = \sum_{\Omega} p(\Omega|s_0)\underbrace{\Big[\log p_\pi(\Omega|s_{f|\Omega}, s_0)p_\rho(\Omega|s_{f|\Omega}, s_0) - \log \pi^p(\Omega|s_0)p_\rho(\Omega|s_0)\Big]}_{r_\Omega^I}\nabla_{\theta_\pi^p} \log \pi^p(\Omega|s_0),$$

$$\nabla_{\theta_\pi^p} I^{VE} = \sum_{\Omega} p(\Omega|s_0)\underbrace{\Big[\log \pi^q(\Omega|s_{f|\Omega}, s_0)\rho^q(\Omega|s_{f|\Omega}, s_0) - \log \pi^p(\Omega|s_0)\rho^p(\Omega|s_0)\Big]}_{r_\Omega^{I^{VE}}}\nabla_{\theta_\pi^p} \log \pi^p(\Omega|s_0),$$

$$(15)$$

where $p_\pi(\Omega|s_0)$ is replaced by $\pi^p(\Omega|s_0)$ since we know the true value of policy (see Appendix A for details). From (15), we see that $\nabla_{\theta_\pi^p} I^{VE} \to \nabla_{\theta_\pi^p} I$ as $\pi^q(\cdot|s_f, s_0) \to p_\pi(\cdot|s_f, s_0)$, $\rho^q(\cdot|s_f, s_0) \to p_\rho(\cdot|s_f, s_0)$ and $\rho^p(\cdot|s_0)) \to p_\rho(\cdot|s_0)$ for all $s_f$. It means that we can estimate the correct gradient of mutual information w.r.t policy as our estimates (10) converge to the true distribution (9). Note that for deterministic environments, we can omit $\rho^q(\cdot|s_{f|.}, s_0)$ and $\rho^p(\cdot|s_0)$ since they are always 1. Then we have $I^{VE} = I^{VB}$ and $\nabla_{\theta_\pi^q} U^{VE} = -\nabla_{\theta_\pi^q} I^{VB}$, which means that maximizing $I^{VB}$ is equivalent to minimizing $U^{VE}$ for $\theta_\pi^q$ (i.e., it is equivalent to the original implicit VIC). Finally, Algorithm 1 summarizes the modified implicit VIC with transitional probability model. The additional steps added to the original implicit VIC are marked with ($*$). Note that Algorithm 1 is not always practically applicable since it is hard to model $p(s_{t+1}|\tau_t, a_t)$ and $p(s_{t+1}|\tau_t, a_t, s_f)$ when the cardinality of the state space is unknown. (We can not set the number of nodes for *softmax*.) In our experiment of Algorithm 1, we will assume that we know the cardinality of the state space. This allows us to model $p_\rho(s_{t+1}|\tau_t, a_t)$ and $p_\rho(s_{t+1}|\tau_t, a_t, s_f)$ using $softmax$. In the next section, we propose a practically applicable method that avoids this intractability of the cardinality.

---

**Algorithm 1** Implicit VIC with transitional probability model

---

Initialize $s_0, \eta, T_{train}, \theta_\pi^q, \theta_\rho^q, \theta_\pi^p$ and $\theta_\rho^p$.
**for** $i_{train} : 1$ to $T_{train}$ **do**
    Follow $\pi^p(a_t|\tau_t)$ result in $\Omega = (s_0, a_0, ..., s_f)$.
    $r_\Omega^{I^{VE}} \leftarrow \sum_t [\log \pi^q(a_t|\tau_t, s_f) - \log \pi^p(a_t|\tau_t)]$                     ▷ from (15)
    $r_\Omega^{I^{VE}} \leftarrow r_\Omega^{I^{VE}} + \sum_t [\log \rho^q(s_{t+1}|\tau_t, a_t, s_f) - \log \rho^p(s_{t+1}|\tau_t, a_t)]$       ▷ from (15), (*)
    **Update each parameter:**
        $\theta_\pi^p \leftarrow \theta_\pi^p + \eta r_\Omega^{I^{VE}} \nabla_{\theta_\pi^p} \sum_t \log \pi^p(a_t|\tau_t)$                   ▷ from (15)
        $\theta_\pi^q \leftarrow \theta_\pi^q + \eta \nabla_{\theta_\pi^q} \sum_t \log \pi^q(a_t|\tau_t, s_f)$                     ▷ from (14)
        $\theta_\rho^p \leftarrow \theta_\rho^p + \eta \nabla_{\theta_\rho^p} \sum_t \log \rho^p(s_{t+1}|\tau_t, a_t)$               ▷ from (14), (*)
        $\theta_\rho^q \leftarrow \theta_\rho^q + \eta \nabla_{\theta_\rho^q} \sum_t \log \rho^q(s_{t+1}|\tau_t, a_t, s_f)$         ▷ from (14), (*)
    **end**
**end for**

---

## 4  IMPLICIT VIC WITH GAUSSIAN MIXTURE MODEL

In this section, we propose the alternative method to overcome the limitation of Algorithm 1 by modeling the smoothed transitional distributions. This allows us to use the Gaussian Mixture Model (GMM) (Pearson, 1894) and other continuous distributional models for modelling transitional distribution. First, we smooth $p(s_{t+1}|\tau_t, a_t, s_f)$ and $p(s_{t+1}|\tau_t, a_t)$ into $f_\sigma(x_{t+1}|\tau_t, a_t, s_f)$ and $f_\sigma(x_{t+1}|\tau_t, a_t)$ with $x_{t+1} = s_{t+1} + z_{t+1}$ and $z_{t+1} \sim \mathcal{N}(\mathbf{0}, \sigma^2 \mathbf{I}_d)$:

$$
\begin{aligned}
f_\sigma(x_{t+1}|\tau_t, a_t, s_f) &= \sum_{s' \in S(\tau_t, a_t, s_f)} p(s'|\tau_t, a_t, s_f) f_\sigma(x_{t+1} - s'; \mathbf{0}, \sigma^2 \mathbf{I}_d), \\
f_\sigma(x_{t+1}|\tau_t, a_t) &= \sum_{s' \in S(\tau_t, a_t)} p(s'|\tau_t, a_t) f_\sigma(x_{t+1} - s'; \mathbf{0}, \sigma^2 \mathbf{I}_d),
\end{aligned}
\tag{16}
$$

where $S(\tau_t, a_t, s_f) = \{s'|p(s'|\tau_t, a_t, s_f) > 0\}$, $S(\tau_t, a_t) = \{s'|p(s'|\tau_t, a_t) > 0\}$ and $d$ is the dimension of the state. Then, using Gaussian Mixture Model (GMM) (Pearson, 1894), we model (16) as $f_\sigma^q(x_{t+1}|\tau_t, a_t, s_f)$ and $f_\sigma^p(x_{t+1}|\tau_t, a_t)$:

$$
\begin{aligned}
f_\sigma^q(x_{t+1}|\tau_t, a_t, s_f) &= \sum_{i=1}^{n_{gmm}} w_i(\tau_t, a_t, s_f) f_\sigma(x_{t+1}; \boldsymbol{\mu}_i(\tau_t, a_t, s_f), \sigma^2 \mathbf{I}_d), \\
f_\sigma^p(x_{t+1}|\tau_t, a_t) &= \sum_{i=1}^{n_{gmm}} w_i(\tau_t, a_t) f_\sigma(x_{t+1}; \boldsymbol{\mu}_i(\tau_t, a_t), \sigma^2 \mathbf{I}_d).
\end{aligned}
\tag{17}
$$

Note that if we set $n_{gmm} > \max_{\tau_t, a_t} |S(\tau_t, a_t)|$, (17) can perfectly fit (16). Now using (17), we define $I_\sigma^{VE}$, the variational estimate with smoothing as follows:

$$
I_\sigma^{VE} = \sum_{\Omega, s_f} p(\Omega, s_f|s_0) \Big[ \log \pi^q(\Omega|s_f, s_0) f_\sigma^q(\Omega|s_f, s_0) - \log \pi^p(\Omega|s_0) f_\sigma^p(\Omega|s_0) \Big].
\tag{18}
$$

Note that $I_\sigma^{VE}$ is not a variational lower bound on $I$, hence we can not also apply derivation of implicit VIC in Gregor et al. (2016). As in Section 3, we start from the absolute difference between $I$ and $I_\sigma^{VE}$. The upper bound on $|I - I_\sigma^{VE}|$ can be obtained as follows:

$$
\left| I - I_\sigma^{VE} \right| \le U_{\sigma,1}^{VE} + U_{\sigma,2}^{VE} \quad \text{with}
$$

$$
U_{\sigma,1}^{VE} = \left| \sum_{\Omega, s_f} p(\Omega, s_f|s_0) \log \left[ \frac{p_\rho(\Omega|s_f, s_0)}{p_\rho(\Omega|s_0)} \frac{f_\sigma^p(\Omega|s_0)}{f_\sigma^q(\Omega|s_f, s_0)} \right] \right|,
\tag{19}
$$

$$
U_{\sigma,2}^{VE} = \sum_{s_f} p(s_f|s_0) D_{\mathrm{KL}} \big[ p_\pi(\cdot|s_f, s_0) p_\rho(\cdot|s_f, s_0) || \pi^q(\cdot|s_f, s_0) p_\rho(\cdot|s_f, s_0) \big].
$$

See Appendix C for the derivation. Note that $U_{\sigma,2}^{VE}$ differs from the first part of $U^{VE}$ in (13). This upper bound implies $U_{\sigma,1}^{VE} \to 0$ as $f_\sigma^q(\cdot|s_f, s_0)/f_\sigma^p(\cdot|s_0) \to p_\rho(\cdot|s_f, s_0)/p_\rho(\cdot|s_0)$ for all $s_f$ and

$U_{\sigma,2}^{VE} \to 0$ if and only if $\pi^q(\cdot|s_f, s_0) \to p_\pi(\cdot|s_f, s_0)$ for all $s_f$ since $U_{\sigma,2}^{VE}$ is the sum of positively weighted KL divergences. To estimate the correct value of mutual information from (18), we minimize $U_{\sigma,1}^{VE}$ and $U_{\sigma,2}^{VE}$. The gradient of $U_{\sigma,2}^{VE}$ can be obtained as below using (3):

$$\nabla_{\theta_\pi^q} U_{\sigma,2}^{VE} = -\sum_\Omega p(\Omega|s_0) \nabla_{\theta_\pi^q} \log \pi^q(\Omega|s_{f|\Omega}, s_0) \tag{20}$$

which can be estimated from the sample mean. As Algorithm 1, it satisfies $\nabla_{\theta_\pi^q} U_{\sigma,2}^{VE} = -\nabla_{\theta_\pi^q} I^{VB}$, which means that minimizing $U_{\sigma,2}^{VE}$ is equivalent to maximizing $I^{VB}$ with respect to $\theta_\pi^q$. Since $U_{\sigma,2}^{VE}$ is 0 if and only if $\pi^q(\cdot|s_f, s_0) = p_\pi(\cdot|s_f, s_0)$ for all $s_f$, this update will make $\pi^q(\cdot|s_f, s_0)$ converge to $p_\pi(\cdot|s_f, s_0)$ for all $s_f$.

Unlike $U_{\sigma,2}^{VE}$, it is difficult to directly minimize $U_{\sigma,1}^{VE}$ due to the absolute value function. However, it can be minimized by estimating the correct ratio between transitional distributions, $p_\rho(\cdot|s_f, s_0)/p_\rho(\cdot|s_0)$, even though each of their true values is unknown. To estimate this ratio, we fit (17) to (16) by minimizing $D_{KL}[f_\sigma(\cdot|\tau_t, a_t, s_f)\|f_\sigma^q(\cdot|\tau_t, a_t, s_f)]$ and $D_{KL}[f_\sigma(\cdot|\tau_t, a_t)\|f_\sigma^p(\cdot|\tau_t, a_t)]$. If $f_\sigma^q$ and $f_\sigma^p$ are parameterized by $\theta_\rho^q$ and $\theta_\rho^p$, the gradients of KL divergences can be obtained as follows:

$$\nabla_{\theta_\rho^q} D_{KL}[f_\sigma(\cdot|\tau_t, a_t, s_f)\|f_\sigma^q(\cdot|\tau_t, a_t, s_f)] = -\int_{x_{t+1}} f_\sigma(x_{t+1}|\tau_t, a_t, s_f) \nabla_{\theta_\rho^q} \log f_\sigma^q(x_{t+1}|\tau_t, a_t, s_f),$$

$$\nabla_{\theta_\rho^p} D_{KL}[f_\sigma(\cdot|\tau_t, a_t)\|f_\sigma^p(\cdot|\tau_t, a_t)] = -\int_{x_{t+1}} f_\sigma(x_{t+1}|\tau_t, a_t) \nabla_{\theta_\rho^p} \log f_\sigma^p(x_{t+1}|\tau_t, a_t),$$

$$\tag{21}$$

which can be estimated from the sample mean. As our estimated smoothed transitional distribution (17) converges to the true smoothed transitional distribution (16), it satisfies $f_\sigma^q(\cdot|s_f, s_0)/f_\sigma^p(\cdot|s_0)$ $\to p_\rho(\cdot|s_f, s_0)/p_\rho(\cdot|s_0)$ which leads to $U_{\sigma,1}^{VE} \to 0$ for finite $\overline{T}$ and $\sigma \ll d_{min}$ where $\overline{T}$ is the average length of trajectories and $d_{min}$ is the minimal distance between two different states (see appendix D for details). Hence, for small enough $\sigma$ and finite length of trajectories, we can minimize $U_{\sigma,1}^{VE}$ to nearly zero. This implies that if we smooth the original transitional distribution with smaller $\sigma$, the smoothed transition will be sharper and the ratio between the transitional probabilities will be estimated more accurately using them. Once we have $(\pi^q(\cdot|s_f, s_0), f_\sigma^q(\cdot|s_f, s_0), f_\sigma^p(\cdot|s_0)) \approx$ $(p_\pi(\cdot|s_f, s_0), f_\sigma(\cdot|s_f, s_0), f_\sigma(\cdot|s_0))$ for all $s_f$ after the update by (20) and (21), we have $I_\sigma^{VE} \approx I$. Now, we can update the policy by obtaining $\nabla_{\theta_\pi^p} I_\sigma^{VE}$ using (3):

$$\nabla_{\theta_\pi^p} I = \sum_\Omega p(\Omega|s_0) \underbrace{\left[ \log \frac{p_\pi(\Omega|s_f, s_0)}{\pi^p(\Omega|s_0)} + \log \frac{p_\rho(\Omega|s_f, s_0)}{p_\rho(\Omega|s_0)} \right]}_{r_\Omega^I} \nabla_{\theta_\pi^p} \log \pi^p(\Omega|s_0),$$

$$\nabla_{\theta_\pi^p} I_\sigma^{VE} = \sum_\Omega p(\Omega|s_0) \underbrace{\left[ \log \frac{\pi^q(\Omega|s_{f|\Omega}, s_0)}{\pi^p(\Omega|s_0)} + \log \frac{f_\sigma^q(\Omega|s_{f|\Omega}, s_0)}{f_\sigma^p(\Omega|s_0)} \right]}_{r_\Omega^{I_\sigma^{VE}}} \nabla_{\theta_\pi^p} \log \pi^p(\Omega|s_0)$$

$$\tag{22}$$

where $\nabla_{\theta_\pi^p} I$ is rewritten from (15). From (22), we see that it satisfies $\nabla_{\theta_\pi^p} I_\sigma^{VE} \to \nabla_{\theta_\pi^p} I$ as $\pi^q(\cdot|s_f, s_0) \to p_\pi(\cdot|s_f, s_0)$ and $f_\sigma^q(\cdot|s_f, s_0)/f_\sigma^p(\cdot|s_0) \to p_\rho(\cdot|s_f, s_0)/p_\rho(\cdot|s_0)$ for all $s_f$, which can be achieved by (20) and (21) as explained previously. Hence, we can estimate the correct gradient of mutual information w.r.t our policy using the estimated smoothed transitional distributions for finite $\overline{T}$ and $\sigma \ll d_{min}$. However, choosing an appropriate value of $\sigma$ is not straightforward since $d_{min}$ and $\overline{T}$ are usually unknown and depend on the environment. Besides, too small $\sigma$ makes the training of $f_\sigma^q$ and $f_\sigma^p$ unstable due to the extreme gradient of Gaussian function near the mean. Another issue of GMM is the choice of a proper $n_{gmm}$ of (17). As explained previously, we can perfectly fit (17) to (16) for $n_{gmm} > \max_{\tau_t, a_t} |S(\tau_t, a_t)|$. We may choose a very large $n_{gmm}$ for the perfect fit but it makes training hard for its complexity. We leave the proper choice of $\sigma$ and $n_{gmm}$ as future works and use empirically chosen values ($\sigma = 0.25$ and $n_{gmm} = 10$) in this paper. Finally, we summarize our method in Algorithm 2. Additional steps added to the original implicit VIC are marked with ($*$).

---

**Algorithm 2** Implicit VIC with Gaussian mixture model

---

Initialize $s_0, \eta, T_{train}, T_{smooth}, \theta_\pi^q, \theta_\rho^q, \theta_\pi^p$ and $\theta_\rho^p$.
**for** $i_{train} : 1$ to $T$ **do**
    Follow $\pi^p(a_t|\tau_t)$ result in $\Omega = (s_0, a_0, ..., s_f)$.
    $r_\Omega^{I_\sigma^{VE}} \leftarrow \sum_t [\log \pi^q(a_t|\tau_t, s_f) - \log \pi^p(a_t|\tau_t)]$            ▷ from (22)
    $r_\Omega^{I_\sigma^{VE}} \leftarrow r_\Omega^{I_\sigma^{VE}} + \sum_t [\log f_\sigma^q(s_{t+1}|\tau_t, a_t, s_f) - \log f_\sigma^p(s_{t+1}|\tau_t, a_t)]$     ▷ from (22), (*)
    **Update each parameter:**
        $\theta_\pi^p \leftarrow \theta_\pi^p + \eta r_\Omega^{I_\sigma^{VE}} \nabla_{\theta_\pi^p} \sum_t \log \pi^p(a_t|\tau_t)$            ▷ from (22)
        $\theta_\pi^q \leftarrow \theta_\pi^q + \eta \nabla_{\theta_\pi^q} \sum_t \log \pi^q(a_t|\tau_t, s_f)$            ▷ from (20)
        $\Delta\theta_\rho^p \leftarrow 0$            ▷ (*)
        $\Delta\theta_\rho^q \leftarrow 0$            ▷ (*)
        **for** $i_{smooth} : 1$ to $T_{smooth}$ **do**            ▷ (*)
            Sample $(z_1, z_2, ..., z_f), z_i \sim \mathcal{N}(\mathbf{0}, \sigma^2 I_n)$
            $\Delta\theta_\rho^p \leftarrow \Delta\theta_\rho^p + \eta \nabla_{\theta_\rho^p} \sum_t \log f_\sigma^p(s_{t+1} + z_{t+1}|\tau_t, a_t)$     ▷ from (21)
            $\Delta\theta_\rho^q \leftarrow \Delta\theta_\rho^q + \eta \nabla_{\theta_\rho^q} \sum_t \log f_\sigma^q(s_{t+1} + z_{t+1}|\tau_t, a_t, s_f)$     ▷ from (21)
        **end for**
        $\theta_\rho^p \leftarrow \theta_\rho^p + \Delta\theta_\rho^p / T_{smooth}$            ▷ (*)
        $\theta_\rho^q \leftarrow \theta_\rho^q + \Delta\theta_\rho^q / T_{smooth}$            ▷ (*)
    **end**
**end for**

---

## 5 EXPERIMENTS

In this section, we evaluate implicit VIC (Gregor et al., 2016), Algorithm 1 and Algorithm 2. We use LSTM (Hochreiter & Schmidhuber, 1997) to encode $\tau_t = (s_0, a_0, ..., s_t)$ into a vector. We conduct experiments on both deterministic and stochastic environments and evaluate results by measuring the mutual information $I$ from samples. To measure $I$, we rewrite (1) using (3) as follows:

$$I(\Omega, s_f|s_0) = -\sum_{\Omega, s_f} p(s_f|s_0) \log p(s_f|s_0) + \sum_{\Omega, s_f} p(\Omega|s_0)p(s_f|\Omega, s_0) \log p(s_f|\Omega, s_0)$$

$$= -\sum_{s_f} p(s_f|s_0) \log p(s_f|s_0)$$

which is maximized when $s_f$ is distributed uniformly. We estimate $\hat{I}$ using the distribution of $s_f$ from the samples, i.e., $\hat{p}(s_f|s_0)$. We apply exponential moving average (0.99 as a smoothing factor) to an average of 5 repetitions for estimating $\hat{p}(s_f|s_0)$. We manually set $T_{max}$ (maximum length of a trajectory) for each experiment such that the termination action is the only available action at $T_{max}$th action. For training GMM of Algorithm 2, the transitional models are trained to predict the distribution of $\Delta x_{t+1} = x_{t+1} - s_t$ instead of $x_{t+1}$ since predicting difference is usually easier than predicting the whole state. For the training, we have a warm-up phase which trains the base function $b(s_0)$ in Gregor et al. (2016) and the transitional models. After the warm-up phase, we update the base function, policy and transitional models simultaneously. Please see Appendix F.1 for details on the hyper-parameter settings.

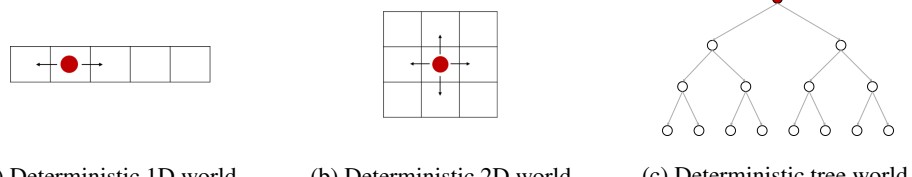

    (a) Deterministic 1D world.        (b) Deterministic 2D world.        (c) Deterministic tree world.

Figure 1: **Deterministic environments.** Fig. 1a shows the deterministic 1D world. The agent can go left right. Fig. 1b shows the deterministic 2D. The agent can go left, up, right and down. Fig 1c shows the deterministic tree world. The agent can go left and right.

We compare the algorithms in deterministic environments in Fig. 1. Note that although (8) is zero for deterministic environments, we still train the transitional models of Algorithm 1 and 2 to show the convergence to the optimum of them. Fig. 2 shows that all three algorithms rapidly achieve the maximal empowerment. We can observer that all the states in environments are visited uniformly after training which is achieved when the mutual information is maximized. Fig. 3 shows the training results of implicit VIC and Algorithm 2 in the $25 \times 25$ grid world with 4 rooms used in Gregor et al. (2016). Both implicit VIC and our Algorithm 2 successfully learn passing narrow doors between rooms and visit almost the whole reachable states for a given $T_{max} = 25$. The additional results in the Mujoco (Todorov et al., 2012) showing the applicability of our Algorithm 2 are in appendix E.

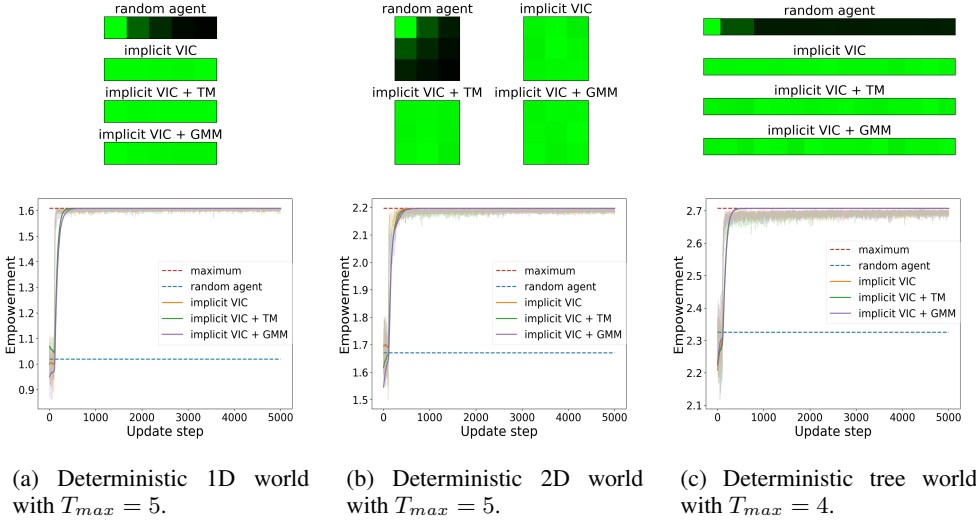

(a) Deterministic 1D world with $T_{max} = 5$.

(b) Deterministic 2D world with $T_{max} = 5$.

(c) Deterministic tree world with $T_{max} = 4$.

Figure 2: **Estimated empowerment during the training in deterministic environments.** Fig. 2a shows the deterministic 1D world and its training results. The agent can go left right. Fig. 2b shows the deterministic 2D world and its training results. The agent can go left, up, right and down. Fig 2c shows the deterministic tree world and its training results. The agent can go left and right. Green shows the distribution of $s_f$.

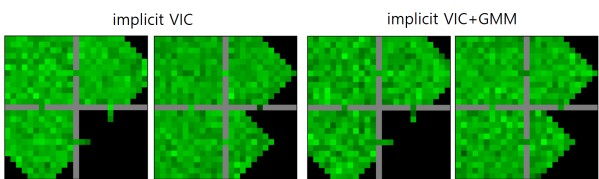

Figure 3: **Deterministic grid world with 4 rooms.** The environment is a $25 \times 25$ grid world with 4 rooms. The agent can go left, up, right and down. The agent starts from (4, 4) and (10, 4) with $T_{max} = 25$. Green shows the distribution of $s_f$.

We compare the algorithms in stochastic environments. Please see appendix F.2 for the details on the stochasticity of the environments. Fig. 4 shows results in simple stochastic environments. We see that while implicit VIC converges to sub-optimum, our two algorithms achieve the maximal empowerment. In Fig. 5, implicit VIC fails to reach far rooms, whereas our Algorithm 2 reaches every room in the environment. From Fig. 4 and 5, we can notice that implicit VIC fails to reach far states under the stochastic dynamics. It happens because the variational bias of implicit VIC is accumulated throughout a trajectory, i.e., it gets larger as the length of the trajectory increases. Fig. 5 also shows the results of training with external rewards. The same mixed reward is used as Gregor et al. (2016), $r = r^I + \alpha r^E$ with $\alpha = 30$. For training the random agent, only $\alpha r^E$ is used and the entropy loss was applied for exploration. While the random agent and implicit VIC converge to sub-optimum, our Algorithm 2 achieves the optimal solution.

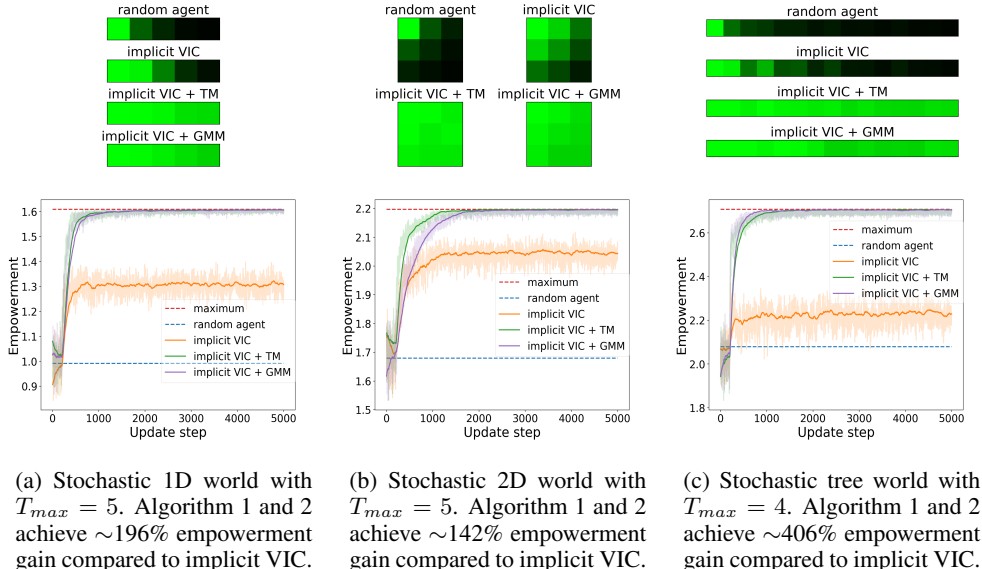

(a) Stochastic 1D world with $T_{max} = 5$. Algorithm 1 and 2 achieve ~196% empowerment gain compared to implicit VIC.

(b) Stochastic 2D world with $T_{max} = 5$. Algorithm 1 and 2 achieve ~142% empowerment gain compared to implicit VIC.

(c) Stochastic tree world with $T_{max} = 4$. Algorithm 1 and 2 achieve ~406% empowerment gain compared to implicit VIC.

Figure 4: **Estimated empowerment during the training in stochastic environments.** The environments are equal to Fig. 1 except for their stochasticity. Green shows the distribution of $s_f$.

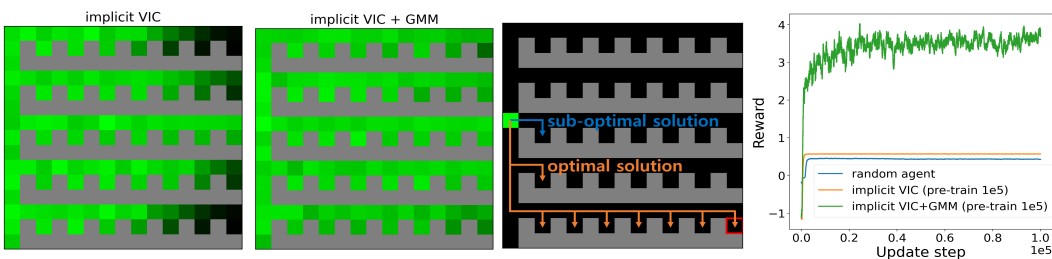

Figure 5: **Stochastic gird world with 35 rooms.** The environment is a $15 \times 15$ grid world with 35 rooms (black cells surrounded by gray walls). We set $T_{max} = 25$. The agent can go up, down and right. It starts from $(0, 6)$. Once the agent enters a room, the environment returns done and then the final action is available only. Green shows the distribution of $s_f$. The external reward is composed of -0.1 for every time step, +1 for entering the normal room and +100 for entering the special room. The sub-optimal solution is reaching the closest room. The optimal solution is trying to reach the special room (the room with the red box) while it enters the closest normal room as soon as possible when it is impossible to reach there due to the stochastic transition.

## 6 CONCLUSION

In this work, we revisited variational intrinsic control (VIC) proposed by Gregor et al. (2016). We showed that for VIC with implicit options, the environmental stochasticity induces a variational bias in the intrinsic reward, leading to convergence to sub-optima. To reduce this bias and achieve maximal empowerment, we proposed to model the environment dynamics using either the transitional probability model or the Gaussian mixture model. Evaluations on stochastic environments demonstrated the superiority of our methods over the original VIC algorithm with implicit options.

ACKNOWLEDGMENTS

This research is conducted with the support of NC. We thank Seong Hun Lee at University of Zaragoza for his sincere feedback on our work, Yujeong Lee at KL Partners for her encouragement and Seungeun Rho at Seoul National University, Jinyun Chung, Yongchan Park, Hyunsoo Park, Sangbin Moon, Inseok Oh, Seongho Son and Minkyu Yang at NC for their useful comments.

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

## A  DERIVATION OF INTRINSIC REWARD FROM MUTUAL INFORMATION

Here we derive the intrinsic reward of VIC by taking the gradient of (1) with respect to the parameter of policy $\theta$. We omit $s_0$ for simplicity. Note that $p(\Omega), p(s_f)$ and $p(\Omega, s_f)$ can be parameterized by $\theta$ since they are all determined by policy. We start by rewriting (1):

$$I(\Omega, s_f) = \sum_{\Omega, s_f} p_\theta(\Omega, s_f) \Big[ \log p_\theta(\Omega, s_f) - \log p_\theta(\Omega) p_\theta(s_f) \Big].$$

Then by taking the gradient with respect to $\theta$, we obtain

$$\nabla_\theta I(\Omega, s_f) = \sum_{\Omega, s_f} p_\theta(\Omega, s_f) \Big[ \log p_\theta(\Omega, s_f) - \log p_\theta(\Omega) p_\theta(s_f) \Big] \nabla_\theta \log p_\theta(\Omega, s_f)$$

$$+ \sum_{\Omega, s_f} p_\theta(\Omega, s_f) \Big[ \frac{\nabla_\theta p_\theta(\Omega, s_f)}{p_\theta(\Omega, s_f)} - \frac{\nabla_\theta p_\theta(\Omega)}{p_\theta(\Omega)} - \frac{\nabla_\theta p_\theta(s_f)}{p_\theta(s_f)} \Big]$$

$$= \sum_{\Omega, s_f} p_\theta(\Omega, s_f) \Big[ \log p_\theta(\Omega, s_f) - \log p_\theta(\Omega) p_\theta(s_f) \Big] \nabla_\theta \log p_\theta(\Omega, s_f)$$

$$+ \sum_{\Omega, s_f} \Big[ \nabla_\theta p_\theta(\Omega, s_f) - p_\theta(s_f|\Omega) \nabla_\theta p_\theta(\Omega) - p_\theta(\Omega|s_f) \nabla_\theta p_\theta(s_f) \Big].$$

Using

$$\sum_{\Omega, s_f} \nabla_\theta p_\theta(\Omega, s_f) = 0, \sum_{\Omega, s_f} p_\theta(s_f|\Omega) \nabla_\theta p_\theta(\Omega) = 0, \sum_{\Omega, s_f} p_\theta(\Omega|s_f) \nabla_\theta p_\theta(s_f) = 0,$$

and (3) we have

$$\nabla_\theta I(\Omega, s_f) = \sum_{\Omega, s_f} p_\theta(\Omega, s_f) \Big[ \log p_\theta(\Omega, s_f) - \log p_\theta(\Omega) p_\theta(s_f) \Big] \nabla_\theta \log p_\theta(\Omega, s_f)$$

$$= \sum_\Omega p_\theta(\Omega) r_\Omega^I \nabla_\theta \log p_\theta(\Omega) \tag{23}$$

where

$$r_\Omega^I = \log p_\theta(\Omega|s_{f|\Omega}) - \log p_\theta(\Omega).$$

Similarly, we can obtain

$$\nabla_\theta I^{VE}(\Omega, s_f) = \sum_{\Omega, s_f} p_\theta(\Omega, s_f) \Big[ \log q(\Omega|s_f) - \log p_\theta(\Omega) \Big] \nabla_\theta \log p_\theta(\Omega, s_f)$$

$$= \sum_\Omega p_\theta(\Omega) r_\Omega^{I^{VE}} \nabla_\theta \log p_\theta(\Omega)$$

where

$$r_\Omega^{I^{VE}} = \log q(\Omega|s_{f|\Omega}) - \log p_\theta(\Omega).$$

Using (4) and (5), we can rewrite $r_\Omega^I$ as

$$r_\Omega^I = \sum_{\tau_t, a_t, s_{t+1} \in \Omega} \log \frac{p_\theta(a_t|\tau_t, s_{f|\Omega}) p_\theta(s_{t+1}|\tau_t, a_t, s_{f|\Omega})}{p_\theta(a_t|\tau_t) p(s_{t+1}|\tau_t, a_t)}.$$

Since $p_\theta(a_t|\tau_t, s_{f|\Omega})$ is intractable, we may replace it with variational inference $q_\phi(a_t|\tau_t, s_{f|\Omega})$ which result in

$$r_\Omega^{I^{VE}} = \sum_{\tau_t, a_t, s_{t+1} \in \Omega} \log \frac{q_\phi(a_t|\tau_t, s_{f|\Omega}) p_\theta(s_{t+1}|\tau_t, a_t, s_{f|\Omega})}{p_\theta(a_t|\tau_t) p(s_{t+1}|\tau_t, a_t)}.$$

For deterministic environment, we have $p_\theta(s_{t+1}|\tau_t, a_t, s_{f|\Omega}) = p(s_{t+1}|\tau_t, a_t) = 1$ and both $r_\Omega^I$ and $r_\Omega^{I^{VE}}$ can be reduced into

$$r_\Omega^I = \sum_{\tau_t, a_t, s_{t+1} \in \Omega} \log \frac{p_\theta(a_t|\tau_t, s_{f|\Omega})}{p_\theta(a_t|\tau_t)}$$

$$r_\Omega^{I^{VE}} = \sum_{\tau_t, a_t, s_{t+1} \in \Omega} \log \frac{q_\phi(a_t|\tau_t, s_{f|\Omega})}{p_\theta(a_t|\tau_t)} = r_\Omega^{I^{VB}}.$$

# B    DERIVATION OF $U^{VE}$

Here we derive $U^{VE}$ from $\left|I - I^{VE}\right|$ with omitted $s_0$ for simplicity:

$$
\left|I - I^{VE}\right| = \left|\sum_{\Omega, s_f} p(\Omega, s_f) \Big[ \log \frac{p_\pi(\Omega|s_f) p_\rho(\Omega|s_f)}{p_\pi^q(\Omega|s_f) p_\rho^q(\Omega|s_f)} - \log \frac{p_\pi(\Omega) p_\rho(\Omega)}{p_\pi^p(\Omega) p_\rho^p(\Omega)} \Big] \right|
$$

$$
\leq \left|\sum_{\Omega, s_f} p(\Omega, s_f) \log \frac{p_\pi(\Omega|s_f) p_\rho(\Omega|s_f)}{p_\pi^q(\Omega|s_f) p_\rho^q(\Omega|s_f)}\right| + \left|\sum_{\Omega, s_f} p(\Omega, s_f) \log \frac{p_\pi(\Omega) p_\rho(\Omega)}{p_\pi^p(\Omega) p_\rho^p(\Omega)}\right|
$$

$$
= \left|\sum_{\Omega, s_f} p(s_f) p(\Omega|s_f) \log \frac{p_\pi(\Omega|s_f) p_\rho(\Omega|s_f)}{p_\pi^q(\Omega|s_f) p_\rho^q(\Omega|s_f)}\right| + \left|\sum_{\Omega, s_f} p(\Omega) p(s_f|\Omega) \log \frac{p_\pi(\Omega) p_\rho(\Omega)}{p_\pi^p(\Omega) p_\rho^p(\Omega)}\right|.
$$

Using (3), (9) and (10), we obtain

$$
\left|I - I^{VE}\right| \leq \left|\sum_{\Omega, s_f} p(s_f) p(\Omega|s_f) \log \frac{p_\pi(\Omega|s_f) p_\rho(\Omega|s_f)}{p_\pi^q(\Omega|s_f) p_\rho^q(\Omega|s_f)}\right| + \left|\sum_{\Omega, s_f} p(\Omega) p(s_f|\Omega) \log \frac{p_\pi(\Omega) p_\rho(\Omega)}{p_\pi^p(\Omega) p_\rho^p(\Omega)}\right|
$$

$$
= \left|\sum_{\Omega, s_f} p(s_f) p(\Omega|s_f) \log \frac{p_\pi(\Omega|s_f) p_\rho(\Omega|s_f)}{p_\pi^q(\Omega|s_f) p_\rho^q(\Omega|s_f)}\right| + \left|\sum_{\Omega} p(\Omega) \log \frac{p_\pi(\Omega) p_\rho(\Omega)}{p_\pi^p(\Omega) p_\rho^p(\Omega)}\right| \ (\because (3))
$$

$$
= \sum_{s_f} p(s_f) D_{\mathrm{KL}}\big[p_\pi(\cdot|s_f) p_\rho(\cdot|s_f) || p_\pi^q(\cdot|s_f) p_\rho^q(\cdot|s_f)\big] + D_{\mathrm{KL}}\big[p_\pi(\cdot) p_\rho(\cdot) || p_\pi^p(\cdot) p_\rho^p(\cdot)\big]
$$

$$
= U^{VE}.
$$

# C    DERIVATION OF $U_{\sigma,1}^{VE}$ AND $U_{\sigma,2}^{VE}$

Here we derive (19). We also omit $s_0$ for simplicity here. We start from $\left|I - I_\sigma^{VE}\right|$:

$$
\left|I - I_\sigma^{VE}\right| = \left|\sum_{\Omega, s_f} p(\Omega, s_f) \Big[ \log \frac{p_\pi(\Omega|s_f) p_\rho(\Omega|s_f)}{p_\pi^q(\Omega|s_f) f_\sigma^q(\Omega|s_f)} - \log \frac{p_\pi(\Omega) p_\rho(\Omega)}{p_\pi^p(\Omega) f_\sigma^p(\Omega)} \Big]\right|
$$

$$
= \left|\sum_{\Omega, s_f} p(\Omega, s_f) \Big[ \log \frac{p_\pi(\Omega|s_f) p_\rho(\Omega|s_f)}{p_\pi^q(\Omega|s_f) f_\sigma^q(\Omega|s_f)} - \log \frac{p_\rho(\Omega)}{f_\sigma^p(\Omega)} \Big]\right| \ (\because p_\pi^p(\Omega) = p_\pi(\Omega))
$$

$$
= \left|\sum_{\Omega, s_f} p(\Omega, s_f) \Big[ \log \frac{p_\rho(\Omega|s_f) f_\sigma^p(\Omega)}{p_\rho(\Omega) f_\sigma^q(\Omega|s_f)} + \log \frac{p_\pi(\Omega|s_f) p_\rho(\Omega|s_f)}{p_\pi^q(\Omega|s_f) p_\rho(\Omega|s_f)} \Big]\right|
$$

$$
\leq \left|\sum_{\Omega, s_f} p(\Omega, s_f) \log \frac{p_\rho(\Omega|s_f) f_\sigma^p(\Omega)}{p_\rho(\Omega) f_\sigma^q(\Omega|s_f)}\right| + \left|\sum_{\Omega, s_f} p(\Omega, s_f) \log \frac{p_\pi(\Omega|s_f) p_\rho(\Omega|s_f)}{p_\pi^q(\Omega|s_f) p_\rho(\Omega|s_f)}\right|
$$

$$
= \left|\sum_{\Omega, s_f} p(\Omega, s_f) \log \frac{p_\rho(\Omega|s_f) f_\sigma^p(\Omega)}{p_\rho(\Omega) f_\sigma^q(\Omega|s_f)}\right| + \left|\sum_{\Omega, s_f} p(s_f) p(\Omega|s_f) \log \frac{p_\pi(\Omega|s_f) p_\rho(\Omega|s_f)}{p_\pi^q(\Omega|s_f) p_\rho(\Omega|s_f)}\right|
$$

$$
= U_{\sigma,1}^{VE} + U_{\sigma,2}^{VE}.
$$

## D   DERIVATION OF THE BOUNDS ON $I - I_\sigma$

Here we derive the bounds on the estimation error of mutual information with smoothing. First, we derive the upper bound on $f_\sigma(s_{t+1}|\tau_t, a_t)$:

$$f_\sigma(s_{t+1}|\tau_t, a_t) = \sum_{s' \in S(\tau_t, a_t)} p(s'|\tau_t, a_t) f_\sigma(s_{t+1} - s'; \mathbf{0}, \sigma^2 \mathbf{I}_n)$$

$$= \frac{1}{\sqrt{(2\pi\sigma^2)^n}} \Big( p(s_{t+1}|\tau_t, a_t) + \sum_{s' \neq s_{t+1} \in S(\tau_t, a_t)} p(s'|\tau_t, a_t) \exp\big(-\frac{\|s_{t+1} - s'\|_2^2}{2\sigma^2}\big) \Big)$$

$$\leq \frac{1}{\sqrt{(2\pi\sigma^2)^n}} \Big( p(s_{t+1}|\tau_t, a_t) + \sum_{s' \neq s_{t+1} \in S(\tau_t, a_t)} p(s'|\tau_t, a_t) \exp\big(-\frac{d_{min}^2}{2\sigma^2}\big) \Big)$$

$$= \frac{1}{\sqrt{(2\pi\sigma^2)^n}} \Big( p(s_{t+1}|\tau_t, a_t) + (1 - p(s_{t+1}|\tau_t, a_t)) \exp\big(-\frac{d_{min}^2}{2\sigma^2}\big) \Big)$$

$$\leq \frac{1}{\sqrt{(2\pi\sigma^2)^n}} \Big( p(s_{t+1}|\tau_t, a_t) + \exp\big(-\frac{d_{min}^2}{2\sigma^2}\big) \Big).$$

Obviously, we have $\frac{1}{\sqrt{(2\pi\sigma^2)^n}} p(s_{t+1}|\tau_t, a_t) \leq f_\sigma(s_{t+1}|\tau_t, a_t)$ which results in:

$$p(s_{t+1}|\tau_t, a_t) \leq \sqrt{(2\pi\sigma^2)^n} f_\sigma(s_{t+1}|\tau_t, a_t) \leq p(s_{t+1}|\tau_t, a_t) + \exp\big(-\frac{d_{min}^2}{2\sigma^2}\big). \qquad (24)$$

Similarly, we can obtain the bounds of $f_\sigma(s_{t+1}|\tau_t, a_t, s_f)$ as follows:

$$p(s_{t+1}|\tau_t, a_t, s_f) \leq \sqrt{(2\pi\sigma^2)^n} f_\sigma(s_{t+1}|\tau_t, a_t, s_f) \leq p(s_{t+1}|\tau_t, a_t, s_f) + \exp\big(-\frac{d_{min}^2}{2\sigma^2}\big). \qquad (25)$$

Combining (24) and (25), we can obtain

$$\frac{p(s_{t+1}|\tau_t, a_t, s_f)}{p(s_{t+1}|\tau_t, a_t) + \exp\big(-\frac{d_{min}^2}{2\sigma^2}\big)} \leq \frac{f_\sigma(s_{t+1}|\tau_t, a_t, s_f)}{f_\sigma(s_{t+1}|\tau_t, a_t)} \leq \frac{p(s_{t+1}|\tau_t, a_t, s_f) + \exp\big(-\frac{d_{min}^2}{2\sigma^2}\big)}{p(s_{t+1}|\tau_t, a_t, s_f)}. \qquad (26)$$

Taking log and using $\log(a + b) \leq \log a + \frac{b}{a}$ for $a, b > 0$ to (26), we have

$$\log \frac{p(s_{t+1}|\tau_t, a_t, s_f)}{p(s_{t+1}|\tau_t, a_t)} - \frac{1}{p_{min}} \exp\big(-\frac{d_{min}^2}{2\sigma^2}\big) \leq$$

$$\log \frac{f_\sigma(s_{t+1}|\tau_t, a_t, s_f)}{f_\sigma(s_{t+1}|\tau_t, a_t)}$$

$$\leq \log \frac{p(s_{t+1}|\tau_t, a_t, s_f)}{p(s_{t+1}|\tau_t, a_t)} + \frac{1}{p_{min,f}} \exp\big(-\frac{d_{min}^2}{2\sigma^2}\big)$$

which results in

$$-\frac{1}{p_{min,f}} \exp\big(-\frac{d_{min}^2}{2\sigma^2}\big) \leq \log \frac{p(s_{t+1}|\tau_t, a_t, s_f)}{p(s_{t+1}|\tau_t, a_t)} - \log \frac{f_\sigma(s_{t+1}|\tau_t, a_t, s_f)}{f_\sigma(s_{t+1}|\tau_t, a_t)} \leq \frac{1}{p_{min}} \exp\big(-\frac{d_{min}^2}{2\sigma^2}\big). \qquad (27)$$

Using (9) and (27) we obtain

$$-\frac{T_\Omega}{p_{min,f}} \exp\big(-\frac{d_{min}^2}{2\sigma^2}\big) \leq \log \frac{p_\rho(\Omega|s_f, s_0)}{p_\rho(\Omega|s_0)} - \log \frac{f_\sigma(\Omega|s_f, s_0)}{f_\sigma(\Omega|s_0)} \leq \frac{T_\Omega}{p_{min}} \exp\big(-\frac{d_{min}^2}{2\sigma^2}\big).$$

By taking the expectation to each side, we have

$$-\frac{T_{max}}{p_{min,f}} \exp\big(-\frac{d_{min}^2}{2\sigma^2}\big) \leq -\frac{\overline{T}}{p_{min,f}} \exp\big(-\frac{d_{min}^2}{2\sigma^2}\big) \leq$$

$$I - I_\sigma \qquad (28)$$

$$\leq \frac{\overline{T}}{p_{min}} \exp\big(-\frac{d_{min}^2}{2\sigma^2}\big) \leq \frac{T_{max}}{p_{min}} \exp\big(-\frac{d_{min}^2}{2\sigma^2}\big)$$

with $T_{max} = \max_\Omega T_\Omega$ and $\overline{T} = \sum_{\Omega, s_f} p(\Omega, s_f|s_0) T_\Omega$. This implies that the estimation error of mutual information with smoothing converges to 0 for fine $T_{max}$ or $\overline{T}$ as $\sigma \to 0$. Also, for finite $\overline{T}$ and $\sigma \ll d_{min}$, it satisfies $|I - I_\sigma| \approx 0$.

## E  ADDITIONAL EXPERIMENT RESULTS

Here we show additional experiment results in HalfCheetah-v3 in the Mujoco environments (Todorov et al., 2012) to show the applicability of our Algorithm 2. We expect that both implicit VIC and our Algorithm 2 will show similar results as Fig. 1 and Fig. 2 since it is a deterministic environment. We fixed the length of the trajectory as $T = 100$ to force the agent to learn long enough trajectories. Each motor's action space is quantized to 5 actions. We can observe that exciting movements (especially triple backflips) are learned by Algorithm 2. Another exciting fact is that since the number of options it can learn is unlimited, it shows newer behaviors on and on as learning goes on.

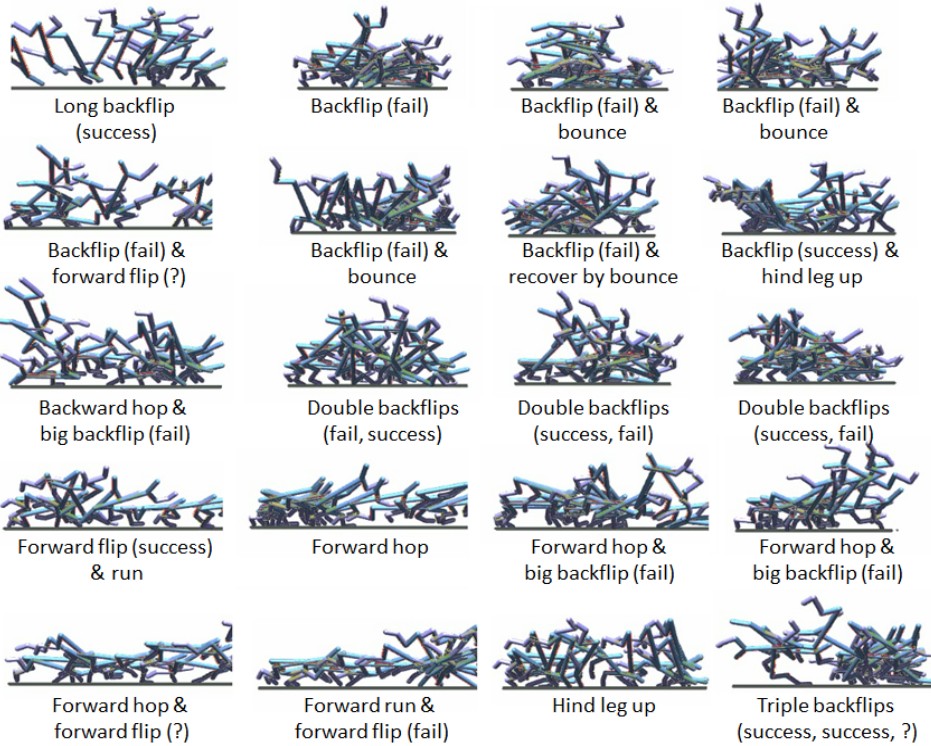

Figure 6: **Learned behaviors by Algorithm 2 in HalfCheetah-v3.** The question mark is written when the result of the flip is unknown. The various behaviors such as forward flip, backflip, etc. are learned by Algorithm 2.

## F  EXPERIMENTAL DETAILS

Here we specify experimental details and environment details.

### F.1  HYPER-PARAMETERS

Here we show hyper-parameters used in our experiments. We used a learning rate of $1e-3$ for Fig. 2 and Fig. 4 and $1e-4$ otherwise.

Table 1: **Hyper-parameters used for experiments**

| Hyper-parameter | Value |
|---|---|
| Optimizer | Adam |
| Learning rate | 1e-3, 1e-4 |
| Betas | (0.9, 0.999) |
| Weight initialization | Gaussian with std. 0.1 and mean 0 |
| Batch size | 128 |
| $T_{smooth}$ | 128 |
| $\sigma$ (GMM) | 0.25 |
| $n_{gmm}$ (GMM) | 10 |

## F.2 STOCHASTIC ENVIRONMENTS DETAIL

Here we specify the transition tables used in this paper.

Table 2: **Transition table of stochastic environments.**

| Transition / Action | go left | go right |
|---|---|---|
| go left | 0.7 | 0.3 |
| go right | 0.3 | 0.7 |

(a) Transition table of the stochastic 1D world.

| Transition / Action | go left | go up | go down | go right |
|---|---|---|---|---|
| go left | 0.7 | 0.1 | 0.1 | 0.1 |
| go up | 0.1 | 0.7 | 0.1 | 0.1 |
| go right | 0.1 | 0.1 | 0.7 | 0.1 |
| go down | 0.1 | 0.1 | 0.1 | 0.7 |

(b) Transition table of the stochastic 2D world.

| Transition / Action | go left | go right | no move |
|---|---|---|---|
| go left | 0.8 | 0.0 | 0.2 |
| go right | 0.6 | 0.2 | 0.2 |

(c) Transition table of the stochastic tree world.

| Transition / Action | go up | go down | go right | no move |
|---|---|---|---|---|
| go up | 0.7 | 0.0 | 0.0 | 0.3 |
| go down | 0.0 | 0.7 | 0.0 | 0.3 |
| go right | 0.0 | 0.0 | 0.7 | 0.3 |

(d) Transition table of the stochastic grid world with 35 rooms.

