# OpenReview forum: "Variational Intrinsic Control Revisited"
_ICLR.cc/2021/Conference — ICLR 2021 Poster_

### Official Review · AnonReviewer3 · 2020-10-19
**Review of "Variational Intrinsic Control Revisited"**

**Rating:** 6
**Confidence:** 4

**Review:**

This paper studies the problem of maximizing empowerment in the context of RL, where the aim is to maximize the mutual information between some latent variable and future outcomes (e.g., future states). The paper first observes that a procedure proposed in prior work [Gregor 16] is biased and hence does not recover a (latent-conditioned) policy that maximizes mutual information. The paper then proposes a new method, based on learning the transition dynamics. that does recover the optimal (mutual information maximizing) policy. Experiments on a few simple tasks show that the proposed method outperforms prior methods.

**Significance**: Empowerment remains one of the main methods for autonomous skill discovery. Thus, a better understanding of how to optimize empowerment would be an important contribution to this area. This paper identifies a limitation (a biased objective) in a commmon formulation of empowerment [Gregor 16], and proposes a method to correct for this. I think the significance of this paper hinges on (1) how large this bias is for reasonably complex tasks, and (2) if this type of bias might occur in other RL objectives, besides empowerment. The paper only convincingly shows that removing this bias is useful for maximizing empowerment on small scale problems.

**Novelty**: To the best of my knowledge, this limitation of VIC has not been discussed in prior work.

**Experiments**
* It would be great to include visualizations or ablation experiments to illustrate why implicit VIC has a lower empowerment than the two proposed methods.
* It'd also be good to include *explicit* VIC as a baseline, even though it requires pre-specifying the number of skills.
* The experiments are limited to very simple gridworlds and tree domains.

**Clarity**
* The derivation of the variational bias in S2 is pretty hard to follow. I'd recommend including a bit more discussion of what implicit VIC is and how it differs from explicit VIC, before continuing with the formal derivation.
* S3 and especially S4 are also hard to follow. I'd recommend moving most of the derivation to the appendix and just stating the final objective as an equation. Theoretical guarantees can then be stated as Theorems/Lemmas with proper proofs.

Overall, I give this paper a score of 5 / 10, primarily because of (1) a lack of clarity and (2) the limited experiments. I would increase my score if the clarify of writing were (greatly) improved, if experiments on higher dimensional tasks were added (e.g., see those in [Achaim 18, Eysenbach 18]), and if additional visualizations of the (suboptimal) behavior of implicit VIC were added.

**Questions for discussion:**
* How significant is the bias in implicit VIC [Gregor] in more complex tasks? Is it significant enough to warrant the additional complexity of the proposed approach?
* Is the GMM approach in S4 just a special case of the model learning approach in S3, where the model is taken to be a GMM?
* Does *explicit* VIC have the same bias as *implicit* VIC?


**Minor comments**
* "methods for measuring it" -- This makes it sound like Arimoto + Blahut proposed methods for measuring empowerment. I'd revise to "along with methods for measuring it based on Expectation Maximization [Arimoto + Blahut]"
* "This can severely degrade the empowerment" -- Clarify what this means.
* "This type of option differs..." -- Aren't there two differences? (1) Closed loop options depend on the state at each time step, and (2) these options include a termination condition.
* " which hinders the maximal level of learning" -- Add a citation.
* "implicit VIC which defines the option as the trajectory until the termination" -- This sentence is confusing without knowing about the method apriori.
* "becomes possible to learn the maximum number of options for the given environment" -- Add a citation.
* "achieve the maximal empowerment" -- It'd be good to formally define what the "maximal empowerment" means.
* "environment dynamics modeling incorporating the transitional probability" -- Grammar error.
* "is the inference to be trained" -> "is the inference network/model to be trained"
* Eq 4: Where are \tau_t and s_{g | \Omega} defined?
* In S3, it might be clearer to use q(...) instead of p^q(...).
* In Eq 15, it's unclear how \log p_\rho^p depends on \theta_\pi^q.
* "when the cardinality of the state space is unknown" -- Where does the *cardinality* of the state space show up as a dependency? Perhaps what is meant is that the method is most readily applicable to *discrete* settings, where the distribution over future states can be approximated exactly, without sampling.
* S4: I'd recommend providing some intuition for why this alternative method is being derived. Is it going to address a limitation of the method in S3?
* "Gaussian Mixture Model (GMM) (Reynolds & Rose, 1995)" -- I think GMMs existed before 1995. E.g., see early work by Karl Pearson.
* "extreme gradient" -- What is an extreme gradient?
* "revisited the variational" -> "revisited variational"

----------------------------------
**Update after author response**:  Thanks to the authors for answering my questions during the rebuttal paper and for incorporating the feedback into the paper! My original concerns were about clarity, high-dimensional experiments, and visualizations. Since the paper has been revised to include nice visualizations and improve the clarity, I am increasing my score 5 -> 6.

I think the experiments on HalfCheetah are a great proof-of-concept of the method! I'd encourage the authors to include some comparisons against baselines for that task.

---

> ### Author Response · Authors · 2020-11-14
> **Response to Reviewer3**
>
> Thanks for your time in reviewing our work. We truly appreciate your comments and would like to improve our work based on your review. We thank you for the clear organization of your feedback on our paper. We would like to respond to your feedback based on your organization.
>
> **Significance**: We agree with the suggested significances. We will explain the significance (1) by answering the questions for discussion. However, it is hard to mention significance (2) by now since it requires more surveys and analyses.
>
> **Experiments**: We will visualize the agent's trajectory after training to show the difference in behavior. We agree that we should expand our experiments to complicated environments and we will try to run an additional experiment for this. We agree that it would be good to include results of explicit VIC and we will try to include the training result of explicit VIC if time allows.
>
> **Clarity**: We agree that our paper is lack of explanation about implicit VIC and especially for explicit VIC. We will explain them in S2 for better understanding. Also, we apologize for the poor readability of S3 and S4. We will try to simplify and re-write our notations for better readability and push more details to appendices.
>
> **Answer to questions for discussion**:
> 1)"How significant is the bias in implicit VIC [Gregor] in more complex tasks? Is it significant enough to warrant the additional complexity of the proposed approach?"
>
> As we can see from equation 8, this bias comes from the difference between transitional probabilities with given and ungiven final state. For bigger bias, their difference should be large and it means that $s_f$ should play a crucial role in the nominator. If $s_f$ does not play a role at all, i.e., $s_f$ is independent of $s_t$, a_t and $s_{t+1}$ then the nominator and the denominator are the same which results in no bias. Large bias happens when $s_{t+1}$ is the necessary state to reach $s_f$ while $s_{t+1}$ is not the only transition from $s_{t}$ and $a_{t}$. In this case, the nominator is $1$ and it generates a large bias. That is why implicit VIC shows severe degradation in our stochastic tree environment. For a given trajectory, all transitions are necessary with the given $s_f$ in this environment. We think that the amount of bias depends on the characteristic of the environment, not the complexity of it. Another example with large bias is the environment with the pairs of keys and doors since a key is necessary to pass a door.
>
> 2)"Is the GMM approach in S4 just a special case of the model learning approach in S3, where the model is taken to be a GMM?"
>
> It is not a special case of S3 and they are different. S3 directly models the transitional probabilities. If we model them using a neural network, it requires the cardinality of state space to set the number of nodes for softmax function.
> The prerequisite of known cardinality is a clear limitation in this case and S4 is suggested to overcome this limitation. If we can model discrete probability without knowing the cardinality of the state space, S3 will be a better solution than S4.
>
> 3)"Does explicit VIC have the same bias as implicit VIC?"
>
> No, explicit VIC does not have the transitional bias since its option does not contain any transition.
>
> **minor comments**: We appreciate all your detailed comments in here. We will improve our paper based on comments. Let us answer questions in minor comments.
>
> 1)"This type of option differs..."
>
> Yes, there are two differences but we neglected closed-loop options since it is mentioned just before this sentence. We will make it clear.
>
> 2)"Eq 4: Where are \tau_t and s_{g | \Omega} defined?"
>
> We use the same definition of $\tau$ as Eq 3 so we omitted it. We will define $\tau_t$ in the text since both Eq 3 and Eq 4 use this.
>
> 3)"In Eq 15, it's unclear how \log p_\rho^p depends on \theta_\pi^q."
>
> We can not find that \log p_\rho^p depends on \theta_\pi^q in Eq.
>
> 4)"Where does the cardinality of the state space show up as a dependency? ..."
>
> The dependency on the cardinality of the state space is explained in 'Answer to questions for discussion - 2)'.
>
> 5)"S4: I'd recommend providing some intuition for why this alternative method is being derived. Is it going to address a limitation of the method in S3?"
>
> The dependency on the cardinality of the state space is a limitation of S3. S4 is suggested to overcome this limitation.
>
> 6)"extreme gradient -- What is an extreme gradient?"
>
> The expression of 'extreme gradient' means that both huge and tiny gradient exists in a very narrow region. For a very small value of standard deviation, the Gaussian distribution function becomes like a delta function which has zero gradient at the mean, infinite gradient around the mean and zero gradient away from the mean. This can cause drastic changes in a small update near mean which results in instability of training.

---

> > ### Comment · AnonReviewer3 · 2020-11-14
> > **Let me know when the paper has been updated**
> >
> > Thanks for the response, and for answering my questions! I think the answers all make sense. I appreciate the willingness to run the additional experiments suggested and to revise the paper to improve clarify. Please let me know when these new pieces have been added, and I'll revisit my review at that time.

---

> > > ### Author Response · Authors · 2020-11-20
> > > **Revised version is uploaded.**
> > >
> > > To Reviewer3.
> > > Thanks to your comments, we have uploaded our revised version.
> > > We will be glad if you check it.
> > > Thanks.

---

> > > > ### Comment · AnonReviewer3 · 2020-11-21
> > > > **Review of revised paper**
> > > >
> > > > Review of revised paper
> > > >
> > > > Thanks for the heads-up! I've just gone through the revised paper.
> > > >
> > > > My original concerns were about clarity, high-dimensional experiments, and visualizations:
> > > > * **clarity**:  I still find the paper pretty hard to read. A few elements that contribute to this are (1) the introduction doesn't mention the problem statement until the final paragraph and (2) Sections 2 - 4 are a "wall of math" without much intuition and discussion about what each part means. For (2), I'd recommend including stating the main results as standalone Theorems/Lemmas and relegating the proofs to the Appendix.
> > > > * **High dimensional experiments**: I didn't see these in the revised paper.
> > > > * **Visualizations of the policies**: The new visualizations in Figures 2/3/4 are really neat! Quick question: In Figure 2 there are _two_ visualizations shown for each method; how do these visualizations differ?
> > > >
> > > > Unless the clarity and high-dimensional concerns are addressed, I plan to stick with my current rating.

---

> > > > > ### Author Response · Authors · 2020-11-22
> > > > > **Response to comments of reviewer3**
> > > > >
> > > > > We appreciate your time in reviewing our revised work.
> > > > >
> > > > > Please let us start by answering the question.
> > > > >
> > > > > **Answer to the quick question**: They are pretty similar and that is what we want to show. Under the deterministic environment, implicit VIC and our algorithms are identical. This figure is for validation that both implicit VIC and our Algorithm 2 works well in the deterministic environment used in Gregor et al. (2016).
> > > > >
> > > > > **Clarity**:
> > > > > * For (1), we agree that the sudden emergence of motivation in the final paragraph makes it unnatural. We will modify it.
> > > > >
> > > > > * For (2), we agree that our paper looks like the "wall of math". However, we believe that now the expressions used in this paper are familiar to RL researchers and one can understand what we aim for by paying attention.  We moved almost derivations to appendices and only the definitions, final results, algorithms are in the main text. (The upper bounds appear almost right after the related definitions.) If we relegate more details to the appendices, we think we are losing our essence inside the main text. The large-scale experimental results can be achieved with various techniques and tuning of the neural network, and computing power but the principles of the bias corrections are the essences of our paper. (1) How do you think of moving the pseudo-code of algorithms to appendices and adding more explanations? (2) How about showing the algorithms with relegating equations 14, 15, 20, 21 and 22?
> > > > >
> > > > > **High dimensional experiments**:
> > > > >
> > > > > * We have conducted our experiments in a larger environment (25$\times$25) with a longer horizon (25) as in Gregor et al. (2016). Since the number of actions is 5 in there, we have $5^{24} \approx 5.96e16$ possible trajectories and the agent needs to learn how to control them for empowerment maximization. We regarded it as a high-dimensional experiment. We think that the dimensionality of an experiment depends on two factors, the dimension of action space and the length of the horizon ($|A|^{H}$). Since it is hard to increase action space (it requires much more work on designing the environment), we increase the length of the horizon instead.
> > > > >
> > > > > * Also, there is a difficulty in running experiments in [Achaim 18, Eysenbach 18]. To show the improvement compared to implicit VIC, we need to control the stochasticity manually. Otherwise, no matter how the environment is complex, Algorithm 2 will show similar results as implicit VIC  if the environment is deterministic. However, changing and adding stochasticity to the simulation is not possible in the 'Mujoco' environments used in [Achaim 18, Eysenbach 18]. In the document of Mujoco, it says "MuJoCo simulations are deterministic with one exception: sensor noise can be generated when this feature is enabled." (http://www.mujoco.org/book/programming.html). We may just run experiments in these environments but it will be the results of implicit VIC which is not our contribution. We want to show our contribution point in the experiment.
> > > > >
> > > > > * If you want to see an experimental result in a more high-dimensional environment, how about showing experiment results with a longer horizon in a larger 2D grid world? Or we may extend it to a 3D grid world. While waiting for your response, we will try to figure out how to modify simulations in Mujoco environments.

---

> > > > > > ### Comment · AnonReviewer3 · 2020-11-24
> > > > > > **Response**
> > > > > >
> > > > > > Apologies for the slightly delayed response.
> > > > > >
> > > > > > **Clarity**: The paper is definitely getting better! I agree that moving Eq 14/15/20/21/22 to the appendix would be good, and would free up a bit of space to add more discussion/intuition. (As a reminder, the revised paper can be 9 pages during the rebuttal period).
> > > > > >
> > > > > > **High dimensional experiments** -- The visualizations of the behaviors learn in Appendix E (Fig 5) look really cool! Are there any quantitative or qualitative comparisons with implicit/explicit VIC? For future reference, one way to make the environment stochastic is to modify the `_get_obs()` method of the environment. Another option is to get the internal joint position (`env.get_state()`), tweak it (e.g., add noise), and then set the state (`env.set_state()`).

---

> > > > > > > ### Author Response · Authors · 2020-11-25
> > > > > > > **Thanks for your response.**
> > > > > > >
> > > > > > > We thank you for your response.
> > > > > > >
> > > > > > > **Clarity**:  We are not sure that we can submit the third revision since the discussion phase is done. Anyway, we will work on our third revision to improve our paper.
> > > > > > >
> > > > > > > **High dimensional experiments**:
> > > > > > > * Thanks for your kind explanation about handling environments. We will check the details based on your information.
> > > > > > > * We don't have a qualitative/quantitative comparison with implicit/explicit VIC since we only ran our Algorithm 2 in HalfCheetah-v3 to show the applicability.
> > > > > > > * Due to the lack of time and resources, we could not run experiments of implicit/explicit VIC. Also, we think the comparison with implicit VIC is not meaningful in a deterministic environment since we already showed their similar results in Fig. 1 and Fig. 2.
> > > > > > > * For qualitative/quantitative comparison, it is hard to measure empowerment in this high dimensional environment as Fig. 1 and Fig. 3. That is why we show qualitative visualization in Fig. 2 and Fig. 4. Also, we can not even visualize the distribution of $s_f$ if the dimension of the state is bigger than 3. We think we may use t-SNE for qualitative visualization in this case.
> > > > > > >
> > > > > > > Thanks.

---

> > > > > ### Author Response · Authors · 2020-11-24
> > > > > **Second revision is uploaded.**
> > > > >
> > > > > Dear reviewer3,
> > > > >
> > > > > To show the applicability of Algorithm 2, we conducted an additional experiment in the Mujoco environment by directly applying our Algorithm 2 to 'HalfCheetah-v3' and its results are in appendix e.
> > > > >
> > > > > Thanks to your comments, we could observe interesting results.
> > > > >
> > > > > We could not modify the simulation to be stochastic to show the difference between implicit VIC and our Algorithm 2 due to its hardness.
> > > > >
> > > > > For the clarity, even though it looks complex by equations, we believe that all the expressions are familiar to RL researchers hence one can understand by paying attention.
> > > > >
> > > > > We think our paper is improved throughout the discussions and we thank you again for your comments and advice.

---

### Official Review · AnonReviewer1 · 2020-10-27
**A weakly motivated extension of VIC with implicit options**

**Rating:** 6
**Confidence:** 4

**Review:**

## Summary

The paper points out a limitation of the implicit option version of the Variational Intrinsic Control (VIC) [Gregor et al., 2016] algorithm in the form of a bias in stochastic environments. Two algorithms are proposed that fix the limitation: the first requiring the size of state space to be known and the second which does not make such assumptions. Experiments on simple discrete state environments demonstrate that the original VIC algorithm works well only on deterministic environments whereas the proposed fix works well on the stochastic environments as well.

## Strengths
- The paper provides a sound theoretical analysis of the limitation of the VIC implicit-option algorithm, the proposed fix and a practical algorithm (*Algorithm 2*).
- A clear distinction is presented with respect to prior work. The differences between the proposed algorithm and VIC shows that the intrinsic reward now has an added term which depends on an approximate model of the transition probability distribution.

## Weaknesses
- The paper focuses on a very specific and narrow topic without providing much stand-alone motivation for the same. It implicitly borrows motivation from prior work (VIC, etc) without providing its own. The rigorous mathematical derivations are simply re-deriving the VIC mutual information bounds with a new added term and with some extra details on how to do it with a gaussian mixture model. Overall, the paper seems like a minor extension of prior work.
- Considering the experiments on partially observed environments presented in the VIC paper, this paper chooses a much simpler set of discrete environments for empirical analysis instead of stepping up to more complicated environments which would have strengthened both the motivation for fixing the bias of VIC and the empirical evidence of the GMM algorithm (*Algorithm 2*).
- The paper's contributions boil down to Eqn 6 (the bias in VIC) and the two proposed algorithms. However, the difficulty in reading the mathematical notations and expressions severely handicaps the reader's ability to carefully understand these contributions. Some suggestions on improving the notation are provided below.

## Feedback to authors
- The paper introduces extremely dense notation. The frequent overloading of symbols or use of similar looking symbols (e.g. $p, p^p, \rho$) makes it quite difficult for the reader to parse each expression. I would recommend usage of longer variable names, e.g.: replace p -> gen, for generative model and replace q -> inf for inference models. Phrases are easier to parse than single character symbols. Also, colorizing certain important symbols can help -- especially for important distinctions such as the true probability distributions vs their estimates.

-------
## Post-rebuttal update

Having read through all reviews and the author's response, I am updating my assessment in light of the responses and new experiments. I agree with the authors that the derivation has theoretical value and is not a simple re-derivation of VIC. The new experiments and visualizations have been helpful (I am happy with the author's responses to R3), but the overall clarity of the paper is still lacking due to the dense mathematical notation. In light of this, I am increasing my score from 4 -> 6, slightly leaning towards acceptance.

---

> ### Author Response · Authors · 2020-11-14
> **Response to Reviewer1**
>
> Thanks for your time in reviewing our work. We truly appreciate your comments and would like to improve our work based on your review. We thank you for the clear summary, strengths, weaknesses and feedback. As you mentioned, our paper is focused on improving the theoretical limitation of Variational Intrinsic Control (VIC) [Gregor et al., 2016] and does not provide our motivation and intuition. We will provide our motivation and intuition in the introduction during the discussion phase. Also, we think that we should run experiments in complicated environments to show the scalability and empirical applicability of Algorithm 2. We will try to run additional experiments for this. We agree that the readability of our paper is poor due to the dense notations and complicated equations. We will try to modify our notations based on your feedback for better readability. Overall, we agree with your comments, however, we want to discuss one weakness mentioned above.
>
> 1)”The rigorous mathematical derivations are simply re-deriving the VIC mutual information bounds with a new added term and with some extra details on how to do it with a gaussian mixture model. Overall, the paper seems like a minor extension of prior work.”
>
> We want to explain that it is not simply re-deriving with the additional term and claim that our derivation is super-set of the one in Gregor et al. (2016) for implicit VIC. The estimate on MI with transitional models is not a variational lower bound on the true MI. This means that simply maximizing estimation with respect to each parameter like Gregor et al. (2016) doesn’t work in this case, hence it requires new and complex analysis on MI. (That is why we express our estimate on MI as ‘variational estimation’ instead of ‘variational bound’.) To tackle this problem, we start with the absolute difference between the true MI and our estimate. However, this absolute symbol can not be removed until applying equation 5 which is only characteristic of implicit VIC, not of explicit VIC (please see appendix B). We want to stress that this is not just simple re-derivation. The reason why we claim that our derivation is a superset of implicit VIC in Gregor et al. (2016) is that our Algorithm 1 is equal to implicit VIC under deterministic dynamics as explained in this paper. It looks like just a simple additional intrinsic reward term and extra updates are added. However, to ensure that they make our estimation more correct and maximize the mutual information under stochastic dynamics,  we believe that complex analysis is inevitable.

---

> ### Author Response · Authors · 2020-11-20
> **Revised version is uploaded.**
>
> To Reviewer1.
> Thanks to your comments, we have uploaded our revised version.
> We will be glad if you check it.
> Thanks.

---

### Official Review · AnonReviewer4 · 2020-10-28
**Needs more experiments and motivation**

**Rating:** 5
**Confidence:** 4

**Review:**

Summary of paper:
The authors study a version of VIC that represents options as partial trajectories. They point out that, in stochastic environments, the implicit VIC formulation in the original paper is missing a term in the mutual information (involving log likelihood ratios of state transitions). They introduce two ways of estimating the missing term: one appropriate for discrete state spaces, and another appropriate for continuous state spaces. They then compare the empowerment of implicit VIC with and without their corrections in a few toy domains, showing that their corrections do not hurt in deterministic environments, and provide a small increase in empowerment in stochastic environments.

Pros:
The missing term highlighted, and the derivations of solutions generally looked correct (at least at the level I followed them). This is a potentially interesting contribution.

Cons:
1) The experiments are a) done only in toy domains and b) even there demonstrate only a 5-10% improvement in empowerment. These experiments are maybe fine as a sanity check, but they are not enough to demonstrate the importance of the authors’ correction term. This is because empowerment is not an important objective in and of itself. Empowerment is used as a unsupervised pre-training step for RL, or as an exploration bonus in conjunction with RL. It is useful to the extent it aids performance in those settings. It is not clear to me whether the correction term is important in those pursuits. It could even be that focussing on the part of empowerment due to stochastic state transitions actually degrades the usefulness of the learnt policy. Additional experiments are needed.

2) In addition to experiments justifying the extra term, it would also be useful for the authors to include more motivation and intuition about why and when it is important. It is not immediately intuitively clear to me why incentivizing an agent to drastically alter its trajectory based on random state transitions is useful.

3) In general, the paper is hard to follow. There are long blocks of equations without enough exposition. New terms are defined without motivating first why they are being introduced. The notation is often too dense (e.g. p^p_p? really?). I mostly felt “in-the-dark” as to where the authors were going while reading the paper. Perhaps they could streamline the derivations, push some of it to the appendix, and spend more time in the main text on motivation and intuition.

4) Given the environments aren’t standard and are very simple, they should definitely be introduced in the main text, and not pushed to the appendix.

---

> ### Author Response · Authors · 2020-11-13
> **Response to Reviewer4**
>
> Thanks for your time in reviewing our work. We truly appreciate your comments and would like to improve our work based on your review. We thank you for the well-organized cons. We generally agree with the above cons but for some part of it, we want to add our explanations. Before we start the discussion, please let us make clear the strength of Algorithm 2. Algorithm 2 alleviates the limitation of Algorithm 1, the prerequisite of the known cardinality of the state space and is not only for continuous state space. (It is also applicable to continuous state space.) We will appreciate you if you recognize this point. Now please let us discuss the cons.
>
> 1)“The experiments are a) done only in toy domains and b) even there demonstrate only a 5-10% improvement in empowerment.”
>
> We want to explain that the empowerment of a random agent (uniform distribution of the policy) is not zero and the improvement of the empowerment by our extension depends on the stochasticity of a given environment. In that sense, the empowerment gain compared to a random agent could be much larger than 5-10% in more stochasticity. From this con, we think that the stochasticities used in this paper were not enough to show meaningful improvement in empowerment. We will increase the stochasticity of the environments, re-run the experiments, and show the empowerment gain compared to a random agent.
>
> 2)“Empowerment is used as a unsupervised pre-training step for RL, or as an exploration bonus in conjunction with RL. It is useful to the extent it aids performance in those settings.”
>
> We agree with this con. We will run additional experiments with external rewards in the same environments after finishing training by VIC.
>
> 3)“In addition to experiments justifying the extra term, it would also be useful for the authors to include more motivation and intuition about why and when it is important.”
>
> We agree with this con. We will try to include more motivation and intuition in the introduction.
>
> 4)“In general, the paper is hard to follow. There are long blocks of equations without enough exposition.”
>
> We apologize for the poor readability of this paper. We will try to improve our notations and write motivations of definitions.
>
> 5)“Given the environments aren’t standard and are very simple, they should definitely be introduced in the main text, and not pushed to the appendix.”
>
> We wanted to put our environmental details in the main text, however, the limit of 8 pages made us unavoidable to push it to the appendix. Since our estimate of MI with transitional models is not a variational lower bound on the true MI as in Variational Intrinsic Control (VIC) [Gregor et al., 2016], it requires new and complex analysis on MI which makes our paper full of dense equations even though we pushed all of the derivations to appendices. We will try to simplify our notations and expressions and we will introduce it in the main text if the page limit allows.

---

> > ### Comment · AnonReviewer4 · 2020-11-19
> > **Let me also know when the paper is updated!**
> >
> > Thanks for your thorough response to myself and the other reviewers - I enjoyed reading all of the responses.
> >
> > 0) I acknowledge your point that Algorithm 2 isn't only for continuous spaces, but rather more specifically removes the need for knowing the cardinality of the state space.
> > 1) A plot of the empowerment gap vs noise level could be useful, for example. This, along with some comment on intuition, would be a helpful first set of toy experiments. Though I emphasize, like the other reviewers, that experiments in more realistic domains are crucial to justifying the importance of this work.
> > 2-4) Please ping me when an updated draft is ready!
> > 5) I acknowledge that conference paper length limitations are often tough on more theory-driven work. But with enough motivation / intuition, you can often push the lengthier parts of derivations to an appendix. As long as the reader is convinced that the result *looks* right, it still makes for a readable paper.

---

> > > ### Author Response · Authors · 2020-11-20
> > > **Revised version is uploaded.**
> > >
> > > To Reviewer4.
> > > Thanks to your comments, we have uploaded our revised version.
> > > We will be glad if you check it.
> > > Thanks.

---

### Official Review · AnonReviewer2 · 2020-11-02
**Technically sound extension to an under-explored algorithm. Additional motivation would help with significance.**

**Rating:** 6
**Confidence:** 4

**Review:**

EDIT: The qualitative results help illustrate what the variational bias entails in practice, and indeed the worse coverage constitutes a problem worth overcoming. The Ant experiment was a good attempt at showing scalability, but the deterministic version isn't terribly informative since then the correction term does nothing. Could add stochasticity by taking some number of random actions between the states the agent sees. I suspect that as you increase stochasticity in this way the uncorrected method would degenerate. Would be a clear accept if you could show that, but as is the paper's contribution is bordering on acceptance. 5-->6

The authors show that implicit VIC is biased in stochastic environment due to its blindness to the effect of its 'option' on the state transition dynamics. This is addressed by learning a model of these dynamics to allow for the calculation of the missing terms. Toy experiments are then performed that show the boost in mutual information caused by eliminating this bias.

First off, its worth mentioning that this is the first real investigation into what intrinsic VIC actually optimizes. The original VIC paper only really explains things for the explicit case, so even the description of what implicit VIC is trying to do is a novel contribution of this work.

That said, the primary merit of implicit VIC was its scalability, and the new requirement of a generative model of state dynamics can only hurt this. For this paper's extension to be truly significant, it should show a case with significant state cardinality (e.g. the 3D environment from the VIC paper) where the bias hurts more than the reliance on the learned model. The GMM approximation suggests this could be possible, there aren't any experiments that really require its usage.

In addition to the scale of the experiments, the breadth of evaluation could be expanded -- showing a gap in the MI is suggestive of a more more interesting gap in behavior. For example, can you use the reverse predictor q(a | s, s_f) to reach all possible states by forcing s_f to equal an arbitrary state? 'Percent of states empirically achievable' should be an easy metric to evaluate in these toy environments and would strengthen the case for your extension.

I have a few minor complaints about the analysis itself. While generally easy to follow, the notation is a bit cumbersome. Once the options are explained in terms of the underlying trajectories, why stick with the option notation? Eliminating the Omegas from the loss terms would be the implications of your extension much for explicit, though perhaps this would just make everything a bit too ugly. I don't follow the significance of equation 5, and would appreciate it being unpacked a bit more. All of the options considered are fixed-length in practice, so I'd omit all of the bits regarding termination actions. What's the difference between showing an upper-bound on the approximation error versus a lower-bound on the true value? If the extension isn't a lower-bound, then the implications of this should be explained.

Overall, I like this paper and wish more papers would be like this. Illuminating a previously neglected algorithm is worthwhile and I want to reward that. But suggesting a model-based approach for an algorithm which cited its model-free nature as a primary motivation requires a more thorough investigation. I'm convinced that you've found a flaw, but I'm not convinced your solution actually improves things.

---

> ### Author Response · Authors · 2020-11-12
> **Response to Reviewer2**
>
> Thanks for your time in reviewing our work. We truly appreciate your comments and would like to improve our work based on your review. We agree that we should run the experiments in high-dimensional environments to show the scalability of Algorithm 2, but for the lack of time and resources, we focused on the main text. However, we will try to run additional experiments in high-dimensional environments during the discussion phase. Also, we will try to simplify the notation for better readability. We agree that the gap in MI may show interesting differences in behavior. We will try to show this difference if time allows. Now please let us discuss about your comments.
>
> 1)"I don't follow the significance of equation 5"
>
> We apologize for the absence of the mention of this significance. This equation is a key characteristic of implicit VIC which is not of explicit VIC. We often use this equation for simple expressions and derivations. The next discussion is also related to the significance of this equation.
>
> 2)"What's the difference between showing an upper-bound on the approximation error versus a lower-bound on the true value? If the extension isn't a lower-bound, then the implications of this should be explained."
>
>  The estimated MI with transitional models is not a lower bound on the true value. So we can not apply the variational lower bound technique used in Variational Intrinsic Control (VIC) [Gregor et al., 2016] and that is why we express as 'variational estimation' instead of 'variational bound' and we start from the absolute difference. For this reason, it requires a complex analysis on MI under stochastic dynamics which makes this paper full of dense expressions.
>  Also, without utilizing equation 5, we can not derive the upper bound in S3. If you see inside Appendix B, right before applying equation 5, the latter part is not a KL divergence, hence the absolute value can not be removed. The latter part can be simplified to KL divergence by using equation 5 and then the absolute value can be removed.
>
> 3)"why stick with the option notation? Eliminating the Omegas from the loss terms would be the implications of your extension much for explicit, though perhaps this would just make everything a bit too ugly."
>
> Since the length of the trajectory varies, using Omega allows us the simplicity of expressions. Otherwise, we need to sum loss over two variables, length of the trajectory and the transitions in the trajectory. We think that the implications of our extension are explicit in Algorithm 1 and Algorithm 2 pseudo-code with the additional term in intrinsic reward.
>
> 4)"All of the options considered are fixed-length in practice, so I'd omit all of the bits regarding termination actions."
>
>  Only the maximum length of the trajectory is fixed in this paper. If we do not have this assumption, an infinite (or very long) length trajectory may occur during the training and it makes the experiment inefficient. Also, we didn't understand 'bits regarding termination actions'. We will appreciate if you explain more detail about this.

---

> > ### Comment · AnonReviewer2 · 2020-11-24
> > **Re: fixed-length trajectories**
> >
> > I misunderstood the trajectory maximum in the paper -- I thought it implied all trajectories were of that length, whereas it appears that it is just a cap on the length. By 'bits regarding termination actions' I meant 'parts of the paper regarding termination actions', and in hindsight I realize this non-technical usage of the word 'bits' is confusing in the context of a paper dealing with information theory!
> >
> > Anyways, I think you sufficiently clarified my concerning about the trajectory length.

---

> > > ### Author Response · Authors · 2020-11-24
> > > **Thanks for your explanation.**
> > >
> > > Dear reviewer2,
> > >
> > > We thank you for your explanation.
> > >
> > > We are glad that your concern is clarified.
> > >
> > > Thanks.

---

> ### Author Response · Authors · 2020-11-20
> **Revised version is uploaded.**
>
> To Reviewer2.
> Thanks to your comments, we have uploaded our revised version.
> We will be glad if you check it.
> Thanks.

---

### Author Response · Authors · 2020-11-21
**Summary of the revision.**

Dear reviewers.

Thanks to your comments, we revised our paper.
Please let us write a summary of our revision for better understanding.

**1)Section 1 (Introduction)**: We added our motivation at the end of the introduction. Some expression based on our intuition (which can not be cited) is removed or modified.

**2)Section 2 (Variational bias)**: We added the additional explanation of explicit VIC and implicit VIC.

**3)Section 3 (Algorithm 1)**: We simplified the notations with some additional explanation about the limitation of Algorithm 1.

**4)Section 4 (Algorithm 2)**: We moved the whole derivation of estimation error on mutual information with GMM smoothing to the appendix.

**5)Section 5 (Experiments)**: We ran additional experiments with the visualization of each algorithm's behavior and our intuitive explanation.

We thank you again for all your comments and very much welcome any feedback.

---

### Author Response · Authors · 2020-11-24
**Summary of the second revision.**

Dear reviewers,

Thanks to your comments, we uploaded our second revised version.

The changes compared to the first revision are as follows:

* **Section 1 (Introduction)**: We updated our motivation to be more clear.
* **Appendix E**: We added an additional experiment result in a high-dimensional environment with a longer length of the option to show the applicability of our Algorithm 2. (Previous maximum length was 25, but we used 100 in the experiment.)

We hope that the result in Appendix E may relieve your concern about the applicability of Algorithm 2.

Thanks.

---

### Decision · Program_Chairs · 2021-01-07
**Final Decision**

**Decision:**

Accept (Poster)

**Comment:**

This paper revisits the under-explored "implicit" variant of Variational Intrinsic Control introduced by Gregor et al. They identify a flaw that biases the original formulation in stochastic environments and propose a fix.

Reviewers agree that there is a [at least a potential, R4] contribution here: "even the description of what implicit VIC is trying to do is a novel contribution of this work", in the words of R2, and "the derivation has theoretical value and is not a simple re-derivation of VIC", in R4's post-rebuttal remarks. Several reviewers raised significant concerns around clarity, which were addressed in an updated manuscript, which also provided new visualizations and new experiments which reviewers found compelling. All reviewers agreed that the revised manuscript was considerably improved.

R4's score stands at the 5, with the other reviewers all standing at 6. R4's main concerns are around whether the missing term in the mutual information identified by the authors is a problem in practice on non-toy tasks (echoing somewhat R3's concerns re: high-dimensional tasks). While this is a valid concern, the function of a conference paper needn't necessarily be to (even attempt) to provide the final word on a matter. Identifying subtle issues such as the one brought forth in this manuscript and re-examining old ideas is a valuable service to the community, and this paper will serve as a beginning to a conversation rather than an end. The AC also considers themselves rather familiar with the original VIC paper, and found the results herein somewhat surprising and noteworthy.

I recommend acceptance, but encourage the authors to incorporate remaining feedback in the camera-ready.